# ABSTRACTIVE SUMMARIZATION THROUGH THE PRISM OF DECODING STRATEGIES

## ABSTRACT

In the realm of natural language generation, abstractive summarization (AS) is at the center of an unparalleled evolution driven by transformer-based language models (LMs). However, the significance of decoding strategies is often neglected, despite their influence on the generated summaries. Given the abundance of token selection heuristics and their accompanying hyperparameters, the community needs directions to steer well-founded decisions based on the task and the target metrics at hand. To fill this gap, we comparatively assess the effectiveness and efficiency of decoding-time techniques for short, long, and multi-document AS. We explore more than 2500 combinations of 3 widely used million-scale autoregressive encoder-decoder models, 6 datasets, and 9 decoding settings. Our findings shed light on the field, demonstrating that optimized decoding choices can yield substantial performance enhancements. In addition to human evaluation, we quantitatively measure effects using 10 automatic metrics, including dimensions such as semantic similarity, factuality, compression, redundancy, and carbon footprint. We introduce PRISM, a pioneering dataset that pairs AS gold input-output examples with LM predictions under a wide array of decoding options.[1]

## 1 INTRODUCTION

Abstractive summarization (AS) is one of the most emblematic and challenging tasks of natural language generation (NLG), aimed at condensing and rephrasing the main gist of textual documents (Sharma & Sharma, 2023). With the advent of transformer-based solutions, autoregressive language models (LMs) have repeatedly demonstrated their prowess in generating human-like summaries (Zhang et al., 2022a). In this red-hot research area, the AS process is typically broken down into two macro-steps: (i) training a neural network to estimate the next-token probability distributions given the input and previously predicted output tokens, (ii) applying an out-of-the-model decoding strategy to control how tokens are selected and strung together at inference time. Drawing parallelism from optics, decoding methods act like a prism: depending on how they are built and tuned with hyperparameters, they reflect model probabilities in different artificial summaries (Figure 1). Therefore, decoding strategies are considered one of the most significant determinants of AS output quality, also responsible for linguistic properties, prediction n-arity, reproducibility, extrinsic hallucinations (van der Poel et al., 2022), and low information coverage (Meister et al., 2022a).

Lamentably, up-to-now AS contributions lean mainly on the conservative use of default decoding settings inherited from existing tools or previous works (Shen et al., 2022). Sometimes, the choices of the decoding algorithm are presented without much discussion (Guo et al., 2022; Zhang et al., 2022b) or are completely omitted (González et al., 2022). The absence of systematic practice in rigorously scrutinizing the impact of decoding raises a natural fear of its underestimation (Gong & Yao, 2023; Ji et al., 2023), fueled, among other things, by the increasing number of heuristics and the complexity of text evaluation (Frisoni et al., 2022a). Thus, researchers urgently demand the release of comprehensive studies to shed light on best decoding practices (Zarrieß et al., 2021). Meister et al. (2022b) have recently demonstrated that decoding methods exhibit task-dependent variations, revealing that broad assertions in favor of one approach over another could lack solid grounding. However, an in-depth examination focused solely on AS is still pending.

---

[1]Code, data, and predictions will be publicly released in case of acceptance (CC-BY-NC-SA 4.0).

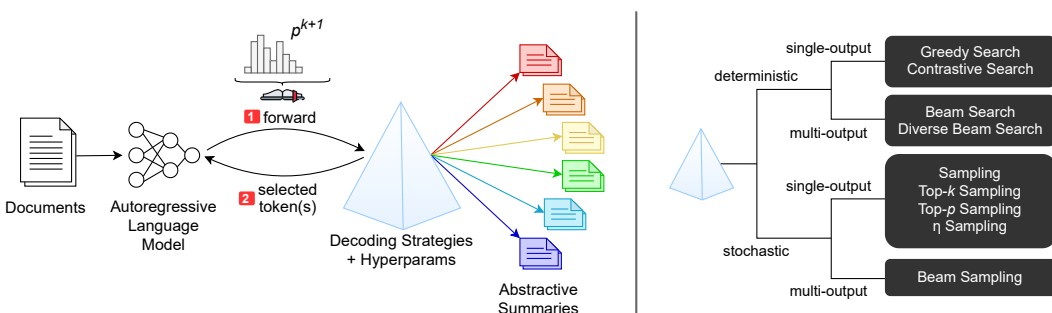

Figure 1: Conceptual division between modeling and decoding in the neural abstractive summarization pipeline (left). Decoding choices determine predicted summaries (🔵). The taxonomy (right) shows the assessed decoding strategies in this work; output n-arity refers to a single generation process.

**Contributions** Our paper fills this lacuna, providing the first side-by-side investigation into the effect of decoding strategies for AS, encompassing short, long, and multi-document settings (see Appendix A for a conceptual preamble). We extensively evaluate 9 well-founded decoding heuristics across 3 state-of-the-art representative encoder-decoder models and 6 widely used datasets from different domains, exploring a broad spectrum of hyperparameters. In addition to human assessment, we put into play 10 distinct automatic evaluation metrics to scan the relationship between decoding and predicted summaries on several quality axes, including naturalness and factuality. Moreover, we judge efficiency by monitoring the carbon footprint and inference time. On balance, this work provides a blueprint for the profitable use of decoding algorithms and helps AS practitioners make confident choices that suit their needs. Our computational dedication shines through the genesis of PRISM, an innovative dataset where gold AS input-output examples are accompanied by a panoply of LM predictions and their decoding metadata. PRISM unleashes novel analysis possibilities, which we posit could assist the community in refining NLG metrics, devising novel decoding strategies, and approximating their inherently non-differentiable behavior.

## 2 RELATED WORK

**Abstractive Summarization** Transformers have assembled a fertile research ground for AS (Kalyan et al., 2021), with efforts to design new large-scale architectures (Chowdhery et al., 2022; Guo et al., 2022), attention mechanisms (Huang et al., 2021a; Phang et al., 2022), and pretraining objectives (He et al., 2022; Wan & Bansal, 2022). Starting in 2019, AS supports a continuous flow of 140+ yearly publications (Figure 2).[2] Many of these works show interest in decoding strategies, referring to them in the title, abstract, or keywords (cf. the blue line in Figure 2). Cross-cutting success has been achieved even in low-resource (Moro & Ragazzi, 2022; 2023; Moro et al., 2023a;b;d) and multi-document (Moro et al., 2022; 2023e) scenarios. The latest trends revolve around the capture of structural properties

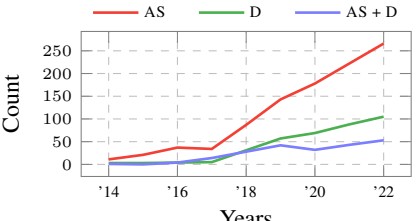

Figure 2: Annual rate SCOPUS comparison between conference papers on AS, decoding (D), and their intersection. See Appendix B for exact queries.

for neural document modeling (Cao & Wang, 2022) or knowledge injection (Frisoni et al., 2022b; 2023), but do not eliminate the need to choose a suitable heuristic. Reinforcement learning sequence-level rewards for direct optimization of NLG metrics have emerged as an alternative to token-level training signals (Ramamurthy et al., 2022; Frisoni et al., 2023). In contrast, this work targets popular million-scale encoder-decoder models trained under maximum likelihood regimes.[3]

---

[2]We direct the interested reader to Syed et al. (2021) and Sharma & Sharma (2023) for a complete overview.

[3]After a wide literature analysis, we prioritized the presently dominant LMs for AS. However, we recognize the lack of exploration in billion-scale networks (15.8% of works since 2020; details are in Appendices B and N). We plan to analyze large LMs, centered on decoder-only architectures and prompting strategies, in future works.

**Decoding Studies** Prior work focused on improving the comprehension of decoding strategies, providing general overviews (Zarrieß et al., 2021), analyzes of human production variability (Giulianelli et al., 2023), or evaluations related to open-ended generation with incomplete optimization processes (Holtzman et al., 2020; Das & Balke, 2022; Su et al., 2022). Studies on AS are often superficial (e.g., only one dataset in single-document settings (Meister et al., 2022b)) or limited to the development of new heuristics without systematic comparisons (Han et al., 2019). To our knowledge, we are the first to (i) thoroughly examine the decoding results for AS in its various forms w.r.t. input length and n-arity (i.e., single and multi-document), (ii) present findings and recommendations based on a literature-supported grid search tuning for decoding hyperparameter spaces, and (iii) release a pioneering decoding-oriented dataset for AS that opens new research directions.

## 3 METHOD

### 3.1 PROBABILISTIC SUMMARY GENERATION

We consider autoregressive encoder-decoder models, with trainable weights $\theta$, that define the conditional probability $p_\theta(\boldsymbol{y}|\boldsymbol{x})$ of a summary $\boldsymbol{y} = \{y_1, \ldots, y_{|\boldsymbol{y}|}\}$ given a variable-length input sequence $\boldsymbol{x}$. Depending on the AS setting, the tokens in $\boldsymbol{x}$ originate from one or more source documents, while the tokens in $\boldsymbol{y}$ are drawn from a finite vocabulary $\mathcal{V}$. The output space of hypothetical summaries is $\mathcal{Y} := \{\text{BOS} \circ \boldsymbol{v} \circ \text{EOS} \mid \boldsymbol{v} \in \mathcal{V}^*\}$, where $\circ$ denotes string concatenation and $\mathcal{V}^*$ Kleene closure of $\mathcal{V}$; valid outputs are enclosed by special tokens "begin-of-sequence" and "end-of-sequence." The models follow a local normalization scheme, factorizing the probability of $\boldsymbol{y}$, as follows:

$$p_\theta(\boldsymbol{y}|\boldsymbol{x}) = \prod_{t=1}^{|\boldsymbol{y}|} p_\theta(y_t \mid \boldsymbol{x}, \boldsymbol{y}_{<t}) \tag{1}$$

where each $p_\theta(\cdot|\boldsymbol{x}, \boldsymbol{y}_{<t})$ is a distribution over $\bar{\mathcal{V}} := \mathcal{V} \cup \{\text{EOS}\}$ and $\boldsymbol{y}_0 := \text{BOS}$. Commonly, weights $\theta$ are learned by minimizing cross-entropy loss $\mathcal{L}(\theta, \mathcal{C})$ between predicted tokens and ground truth ones in a training corpus $\mathcal{C}$ (negative log-likelihood under $p$). To penalize overconfident output and combat overfitting, $\mathcal{L}$ can be enhanced by label smoothing, discounting a certain probability mass of the true token, and redistributing it uniformly across all other tokens (Gao et al., 2020). The purpose of decoding is to find the most probable summary among all candidate hypotheses, i.e., $\boldsymbol{y}^* = \operatorname{argmax}_{\boldsymbol{y} \in \mathcal{Y}} p_\theta(\boldsymbol{y}|\boldsymbol{x})$. This optimization problem is known as the maximum a posteriori. Since the number of possible summaries in the symbolic space increases as $|\mathcal{V}|^{|\boldsymbol{y}|}$, the exact search is NP-hard. Furthermore, even if an exact solution were tractable, it would be far from high quality text (Eikema & Aziz, 2020). Thus, decoding is exclusively approximated with heuristic methods.

### 3.2 DECODING STRATEGIES

We test the full inventory of heuristics supported by HuggingFace as of April 2023,[4] except those aimed at forcing or excluding the generation of specific tokens (out of our AS scope). Such algorithms can be set up to satisfy different NLG needs and fall into two predominant categories: *deterministic* and *stochastic*. The first optimizes for summary continuations with high probabilities, whereas the second puts randomness in place. Table 1 shows our decoding landscape. Note that all algorithms end when $y_t$ or the hypotheses in $\mathcal{Y}_t$ reach EOS for some $t < max\_length$. We do not consider other early stopping rules because they generally do not affect generation quality (Meister et al., 2020).

## 4 EXPERIMENTAL SETUP

### 4.1 EXPERIMENT GOALS

We aim to answer the following research questions. Q1 Is there an absolute best decoding method for AS? Q2 Are decoding methods sensitive to AS type? Q3 To what extent does proper decoding affect LM metrics? Q4 Which decoding method provides the best effectiveness–efficiency trade-off? Q5 Which hyperparameter values best suit a particular AS quality attribute?

---

[4]https://huggingface.co/docs/transformers/internal/generation_utils

Table 1: A summary of the decoding strategies benchmarked in this abstractive summarization study.

| DECODING STRATEGIES[*,†] | | |
|---|---|---|
| **Greedy Search** | **Contrastive Search** (Su et al., 2022) | **Beam Search** (Lowerre, 1976) |
| It selects the most probable token at each $t$ (locally-optimal decision): $$y_t = \underset{y \in \bar{\mathcal{V}}}{\arg\max} \log p_\theta(y \mid \boldsymbol{x}, \boldsymbol{y}_{<t}) \quad \text{(for } t > 0)$$ | It extends Greedy Search by jointly considering the model confidence (i.e., semantic coherence) and the similarity w.r.t. the previous context: $$y_t = \underset{y \in \bar{\mathcal{V}}^{(k)}}{\arg\max} \Big\{ (1-\alpha) \times p_\theta(y\|\boldsymbol{x}, \boldsymbol{y}_{<t})$$ $$-\alpha \times (\max\{s(h_y, h_{y_j}) : 1 \le j \le t-1\}) \Big\}$$ (for $t > 0$) $\bar{\mathcal{V}}^{(k)}$ is the set of top-$k$ predictions from the LM's probability distribution ($k \in \mathbb{Z}_+$). The second term is a degeneration penalty governed by $\alpha \in [0,1]$; it is computed with the cosine similarity $s(\cdot, \cdot)$ between the candidate representation $h_y$ and the prior tokens. When $\alpha = 0$, Contrastive Search degenerates to Greedy Search. | A pruned breadth-first search algorithm that expands the hypotheses in a greedy left-to-right way, retaining the top-$b$ candidates at each $t$: $$Y_t = \underset{\substack{Y' \subseteq \mathcal{B}_t, \\ \|Y'\|=b}}{\arg\max} \mathcal{S}(Y'_t) \quad \text{(for } t > 0)$$ $\mathcal{B}_t$ denotes all possible roll-out extensions $\{\boldsymbol{y}_{t-1} \circ y \mid y \in \bar{\mathcal{V}} \text{ and } \boldsymbol{y}_{t-1} \in Y_{t-1}\}$. $\mathcal{S} : \mathcal{Y} \to \mathbb{R}$ is a scoring function on $Y \subseteq \mathcal{Y}$. By default, $\mathcal{S}(Y) = \sum_{\boldsymbol{y} \in Y} \log p_\theta(\boldsymbol{y}\|\boldsymbol{x})$. $\boldsymbol{y}*$ is chosen from the final set $Y_T$. |
| **Diverse Beam Search** (Vijayakumar et al., 2018) | **Sampling** | **Top-$k$ Sampling** (Fan et al., 2018) |
| In vanilla Beam Search, as $t$ and $b$ increase, the candidate summaries of a single beam gradually occupy the top positions. This inner functioning results in a high overlap among hypotheses that often only differ by punctuation and slight morphological variations (Han et al., 2019), indicating poor search space coverage. Diverse Beam Search splits a beam into sub-groups sequentially optimized and adds a penalty in $\mathcal{S}$ to encourage inter-group dissimilarity: $$\mathcal{S}(Y_t^{(g)}) = \sum_{\boldsymbol{y} \in Y} \log p_\theta(\boldsymbol{y}\|\boldsymbol{x})$$ $$-\lambda \sum_{g' < g} \Delta\left(\boldsymbol{y}, Y_t^{(g')}\right)$$ $\Delta(\boldsymbol{y}, Y_t^{(g')})$ is a diversity measure whose strength is controlled by $\lambda$, and $g$ is a sub-group. | Instead of approximating $\boldsymbol{y}^*$, it makes a random selection at each $t$, thus giving a non-zero chance to every token following the model distribution ($\sim$): $$y_t \sim p_\theta(\cdot\|\boldsymbol{x}, \boldsymbol{y}_{<t}) \quad \text{(for } t > 0)$$ Softmax can be re-calibrated through a temperature parameter $\tau \in (0, 1]$. By decreasing $\tau$ (1 = no effect), we skew the distribution towards likely tokens and reduce the mass in the unreliable tail: $$p_\theta(y\|\boldsymbol{x}, \boldsymbol{y}_{<t}) = \frac{\exp(\frac{u}{\tau})}{\sum_j \exp(\frac{u_j}{\tau})}$$ where $u \in U$ are logits. | It limits the sampling space to the top-$k$ probable tokens, regardless of the distribution shape. Here, the decoding maximizes: $$\sum_{y \in \bar{\mathcal{V}}^{(k)}} p_\theta(y\|\boldsymbol{x}, \boldsymbol{y}_{<t})$$ where $\bar{V}^{(k)} \subseteq \bar{V}$. |
| **Top-$p$ (Nucleus) Sampling** (Holtzman et al., 2020) | **Beam Sampling** (Caccia et al., 2020) | **$\eta$ Sampling** (Hewitt et al., 2022) |
| Instead of assuming a fixed-sized shortlist, it dynamically expands and contracts the candidate pool according to the shape of the distribution (e.g., fewer contenders if sharp). When many next tokens are plausible, the allowed set reflects that. Formally, it samples from the smallest subset of tokens whose cumulative probability mass exceeds a chosen threshold $p \in (0, 1]$: $$\sum_{y \in \bar{\mathcal{V}}^{(p)}} p_\theta(y\|\boldsymbol{x}, \boldsymbol{y}_{<t}) \ge p$$ | It is a variant of Beam Search where, at each $t$, the $b$ next tokens for constructing $\mathcal{B}$ are sampled conditioned on the current hypotheses (no local optimum). | It samples tokens above an entropy-dependent probability threshold to avoid unnecessary cutting-offs. Eligible tokens come from: $$\{y \in \bar{\mathcal{V}} \mid p_\theta(y\|\boldsymbol{x}, \boldsymbol{y}_{<t}) > \eta\}$$ $$\eta = \min\left(\epsilon, \alpha \exp(-e_{\boldsymbol{x},\boldsymbol{y}_{<t}})\right)$$ where $\exp(-e_{\boldsymbol{x},\boldsymbol{y}_{<t}})$ is the expected next token probability given the entropy $e_{\boldsymbol{x},\boldsymbol{y}_{<t}}$, scaled by a constant $\alpha$. To expose a single hyperparameter, $\alpha$ is set to $\sqrt{\epsilon}$. The general principle is to only truncate tokens whose probabilities are low relative to the rest of the distribution or an absolute threshold. |

[*] Basic notation. $\boldsymbol{x}$ = document(s), $\boldsymbol{y}$ = summary, $\theta$ = model weights, $t$ = decoding step, $\bar{\mathcal{V}} := \mathcal{V} \cup \{\text{EOS}\}$, $p_\theta$ = model probability distribution, $\mathcal{Y}$ = hypothetical summaries, $b$ = beam size, $\boldsymbol{y}*$ = best summary.

[†] • = deterministic, • = stochastic.

We plan a battery of generative experiments $(S_{\mathcal{H}}, \mathcal{D}, \mathcal{M})$, where $S_{\mathcal{H}}$ denotes a decoding strategy with fixed hyperparameters, $\mathcal{D}$ is the dataset that imposes particular AS challenges and length constraints, and $\mathcal{M}$ is the autoregressive LM. In total, we executed 2656 runs (Table 2) for 73 GPU days. As far as we know, we drive the finest-grained hyperparameter sweep in the NLG literature. Specifically, we use grid search to simultaneously alter the generative variables of each heuristic, wisely choosing range values based on hints given in previous studies. See Appendices D and E for more details.

## 4.2 DATASETS

We consider 6 noteworthy, English-language, domain-distinguished, and publicly available datasets as testbeds, 2 for each AS family. Preprocessing is carefully applied (Appendix C).

- *Short Document Summarization (SDS)*. **XSUM** (Narayan et al., 2018), a dataset of BBC articles from 2010-2017, designed for highly abstractive one-sentence extreme summarization.

Table 2: Hyperparameters for each decoding strategy.

| HYPERPARAMS[*,†] | Deterministic | | | | Stochastic | | | | |
|---|---|---|---|---|---|---|---|---|---|
| | Greedy | Contrastive | Beam Search | Diverse Beam Search | Sampling | Top-$k$ Sampling | Top-$p$ Sampling | Beam Sampling | $\eta$ Sampling |
| no_rep_ngram_size | | | | | { 2, 3, 4, 5 } | | | | |
| early_stopping | | | ✓ | ✓ | | | | ✓ | |
| diversity_penalty | | | | 0.{2, 4, 6, 8}, 1.0 | | | | | |
| num_beams | | | [2, 3, …, 10] | [2, 3, …, 10] | | | | [2, 3, …, 10] | |
| temperature | | | | | 0.{8, 9}, 1.0 | 0.{8, 9}, 1.0 | 0.{8, 9}, 1.0 | | |
| top_k | | [20, 30, …, 60] | | | | [20, 30, …, 60] | {/, 20, …, 60} | | |
| top_p | | | | | | | 0.{4, 6, 8} | 0.9 | |
| penalty_alpha | | 0.{2, 4, 6, 8}, 1.0 | | | | | | | |
| eta_cutoff | | | | | | | | | {4, 2}×10⁻³, {9, 6, 3}×10⁻⁴ |
| do_sample | | | | | ✓ | ✓ | ✓ | ✓ | ✓ |
| # Runs | 16 | 400 | 144 | 720 | 48 | 240 | 864 | 144 | 80 |

[*] {. . .} = sets, [. . .] = series, ✓ = true, / = none.
[†] orange and cyan initialization schemes are tested on 2 and 4 datasets, respectively; all other values are equally applied to all the 6 surveyed datasets.

**CNN/DM** (Nallapati et al., 2016), a set of articles from different news outlets accompanied by short-sentence highlights written by journalists.

- *Long Document Summarization (LDS)*. **PUBMED** and **ARXIV** (Cohan et al., 2018), two datasets of lengthy and structured scientific papers along with their abstracts.
- *Multi-Document Summarization (MDS)*. **MULTI-NEWS** (Fabbri et al., 2019), composed of news articles and human-written summaries from `newser.com`, which brings together hundreds of US and international sources. **MULTI-LEXSUM** (Shen et al., 2022), a collection of expert-authored summaries of court documents from federal civil rights lawsuits.[5]

**Data Sampling** Given the huge research space for decoding and evaluation, it is impractical to work with entire datasets due to time and resource limitations. Thus, we opt for a representative subset of each dataset, leaving out the compact MULTI-LEXSUM. We use power analysis to estimate the minimum sample size required to detect statistically significant metric effects (Appendix C). We select a 10% dataset size by performing proportional stratified random sampling without replacement to guarantee the adequate representation of each subgroup. We stratify based on the document and summary length; tertiles are calculated to assign {short, medium, large} classes. For MDS, we also consider the number of source documents. A compendium of the datasets is given in Appendix C.

### 4.3 MODELS

To ensure fairness, we use 3 comparable state-of-the-art transformer-based models in their large version for which original fine-tuned weights on the datasets in Section 4.2 already exist. **BART** (Lewis et al., 2020) (SDS, 406M parameters) has quadratic memory complexity in input size, limited to elaborate sequences up to 1024 tokens. **LED** (Beltagy et al., 2020) (LDS, 447M) uses sparse attention to endow BART with a linear input scale, processing up to 16,384 tokens. **PRIMERA** (Xiao et al., 2022) (MDS, 447M) adapts LED to multi-inputs through a summarization-specific pretraining objective, concatenating the sources with a special token and forming a single input of up to 4096 tokens.

### 4.4 EVALUATION

**Automatic** We conjecture that AS quality estimation is similar in complexity to correctly performing the task. On the path of good practice, we take advantage of a panoply of metrics that capture separate attributes (Table 3). We share hyperparameters and takeaways in Appendix E.

**Human** In order to better gauge the merits of the decoding, we conduct a meticulous human evaluation. Motivated by Narayan et al. (2018), Fabbri et al. (2019), and Huang et al. (2023), we use a direct comparison strategy that has been shown to be more reliable, sensitive, and less labor-intensive than rating scales. We sample 5 documents from every dataset. For each, three English-proficient AS researchers are presented with summaries inferred by 2 out of 3 sources, i.e., the top-3 decoding strategies on the dataset (optimal hyperparameter settings according to normalized average scores of effectiveness-oriented metrics, i.e., $\mathcal{R}$, BERTScore, BARTScore). We ask reviewers to select the

---

[5]Given the multi-granularity nature of summaries in MULTI-LEXSUM, we contemplate the D → S task, namely synthesizing the source documents into a *short* summary.

Table 3: A summary of the metrics used in this study to assess the generated summaries.

| METRICS[*,†] | | |
|---|---|---|
| **ROUGE** (Lin, 2004) $[0, 1], \uparrow$ | **BERTScore** (Zhang et al., 2020b) $[-1, 1], \uparrow$ | **Perplexity** (Jelinek et al., 1977) $[0, \infty[, \downarrow$ |
| Unigrams $(r_1)$, bigrams $(r_2)$, and longest common subsequence $(r_{L\text{sum}})$ lexical overlaps (%) between the inferred and gold summaries, i.e., a proxy for informativeness and fluency. Inspired by Moro et al. (2023c), we measure an aggregated judgment: $$\mathcal{R} = \frac{\text{avg}(r_1, r_2, r_{L\text{sum}})}{1 + \sigma_r^2} \quad (2)$$ where $\sigma_r^2$ is the variance of 0-1 normalized ROUGE F1 scores. $\mathcal{R}$ penalizes model results with discrepant $r_1, r_2, r_{L\text{sum}}$ values. | Semantic recall formalized as: $$\frac{\sum_{y_i \in \boldsymbol{y}} \text{idf}(y_i) \max_{\hat{y}_j \in \hat{\boldsymbol{y}}} \text{e}(y_i)^\mathsf{T} \text{e}(y_j)}{\sum_{y_i \in \boldsymbol{y}} \text{idf}(y_i)}$$ where $\text{e}(\cdot)$ is a BERT (Devlin et al., 2019) token embedding. | The naturalness of a summary w.r.t. the data seen by a model (GPT-2 (Radford et al., 2019) in this paper) during training: $$ppl(\boldsymbol{y}) = \exp\left(\frac{1}{t}\sum_i^t \log p_\theta(y_i|\boldsymbol{y}_{<i})\right)$$ . |
| **Coverage** (Grusky et al., 2018) $[0, 1], \uparrow$ | **Density** (Grusky et al., 2018) $[0, |x|_c], \downarrow$ | **Compression** (Grusky et al., 2018) $[0, |x|], \uparrow$ |
| The percentage of summary words within the source text: $$\frac{1}{|\boldsymbol{y}|} \sum_{\boldsymbol{f} \in \mathcal{F}(\boldsymbol{x}, \boldsymbol{y})} |\boldsymbol{f}|$$ where $\mathcal{F}$ is the set of all fragments, i.e., extractive character sequences. When low, it suggests a high chance for unsupported entities and facts. | The average length of the extractive fragments. It is formulated as: $$\frac{1}{|\boldsymbol{y}|_c} \sum_{\boldsymbol{f} \in \mathcal{F}(\boldsymbol{x}, \boldsymbol{y})} |\boldsymbol{f}|_c^2$$ where $||_c$ is the character length. When low, it suggests that most summary sentences are not verbatim extractions from the sources (abstractive). | The document-summary word ratio: $|\boldsymbol{x}|/|\boldsymbol{y}|$. |
| **Unique N-gram Ratio (UNR)** (Xiao & Carenini, 2020b) $[0, 1], \uparrow$ | **Normalized Inverse of Diversity (NID)** (Xiao & Carenini, 2020b) $[0, 1], \downarrow$ | **BARTScore-F** (Yuan et al., 2021) $[-\infty, 0], \uparrow$ |
| The summary $n$-grams uniqueness: $$\frac{\text{count}(\text{uniq\_n\_gram}(\boldsymbol{y}))}{\text{count}(\text{n\_gram}(\boldsymbol{y}))}$$ where we take $n$ from $[1, 3]$ and divide the average by variance. | It reckons redundancy by inverting the entropy of summary unigrams and applying length normalization: $$1 - \text{entropy}(\boldsymbol{y})/\log(|\boldsymbol{y}|)$$ | The weights $\theta$ of a pretrained BART model (Lewis et al., 2020) are used to estimate how likely hypothesis and reference are reciprocal paraphrases (i.e., probability of generating one giving the other): $$\sum_{t=1}^{|\boldsymbol{y}|} \log p(y_t|\boldsymbol{y}_{<t}, \boldsymbol{x}, \theta_B)$$ We measure faithfulness by mapping sources and predicted summaries to $\boldsymbol{x}$ and $\boldsymbol{y}$. |
| **Carburacy** (Moro et al., 2023c) $[0, 1], \uparrow$ | | |
| Carbon-aware accuracy measure modeling both the AS model effectiveness and eco-sustainability: $\Upsilon = \left(\exp^{\log_\alpha \mathcal{R}}\right)/1 + \mathcal{C} \cdot \beta$ where $\mathcal{R}$ is defined in Equation 2, $\mathcal{C}$ is the kg of $CO_2$ emissions produced by the model to process a single instance $x$ at inference time, $\alpha$ and $\beta$ are trade-off hyperparameters. | | |

[*] Blue text indicates the bound value and the general reading key (i.e., $\uparrow$ = higher is better, $\downarrow$ = lower is better).

[†] ● = reference-based (gold-summary dependence), ● = reference-free

better one w.r.t. 4 dimensions; a "tie" is declared if a judge perceives the two summaries to be of equal quality. When considering all possible combinations of summary pairs, the total number of preference labels per participant is 540. We randomize the order of pairs and summaries per example to guard the rating against being gamed. Zooming in, the rating axes are defined as follows. *Recall* considers whether the generated summary covers all target content. *Precision* checks if the generated summary covers only the target content (i.e., no redundant or superfluous information). *Faithfulness* examines whether the generated summary is factually consistent with the input document. *Fluency* dissects the grammaticality and coherence of the summaries. The final score of each decoding strategy is the percentage of times that its summaries are selected as the best, minus the percentage of times that they are not. Appendix F illustrates our setup with human instructions.

## 5 RESULTS

The core results (averaged by the decoding method across all inference runs) are plotted in Figure 3; see Appendix G for tabulated data. Please note that the enumerated quantitative findings are established on metrics that serve as proxies for the quality dimensions of interest. Input-output examples for all decoding strategies are illustrated in Appendix M.

### 5.1 QUANTITATIVE FINDINGS

- A1 **There is no one-size-fits-all strategy for AS.** No single decoding method consistently achieves the highest quality score in all evaluation metrics. However, considering the mean average ranking in terms of $\mathcal{R}$ (the de facto standard AS metric), deterministic approaches secure the first position in every dataset. Properly calibrated stochastic methods perform

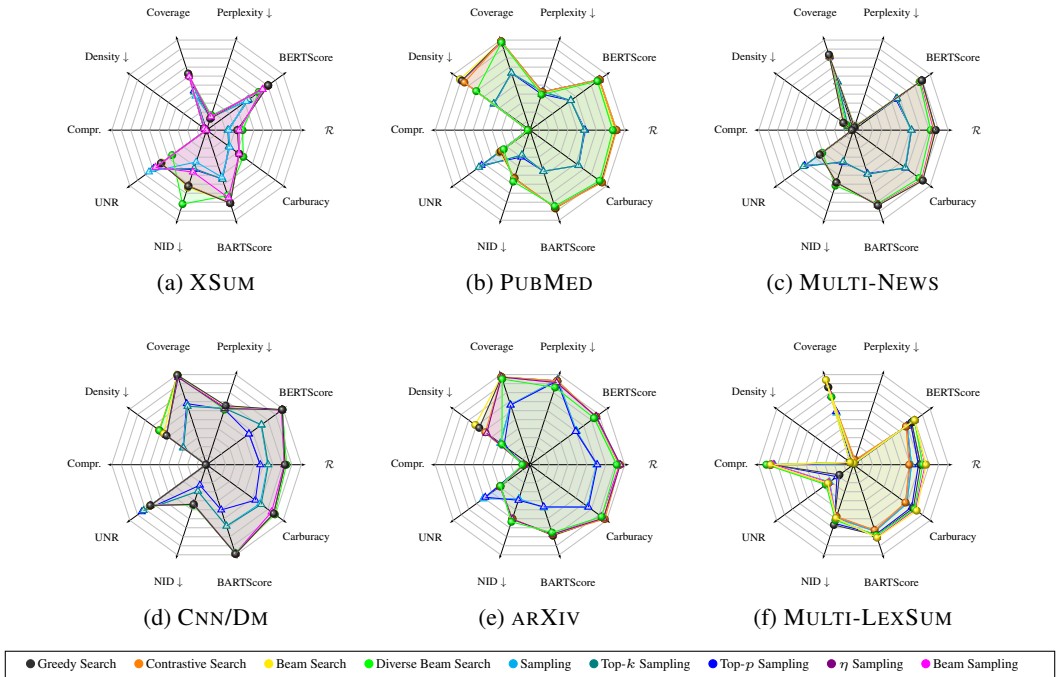

Figure 3: Metric-based comparison between artificial and gold summaries in the examined datasets. Colored areas signify the normalized average scores of different decoding strategies ($\bullet$ = deterministic, $\blacktriangle$ = stochastic). The dominant color in each dataset-specific radar denotes the best overall strategy. Succeeding Cao et al. (2022), $[0, 1]$ rescaling is based on min-max normalization across all runs.

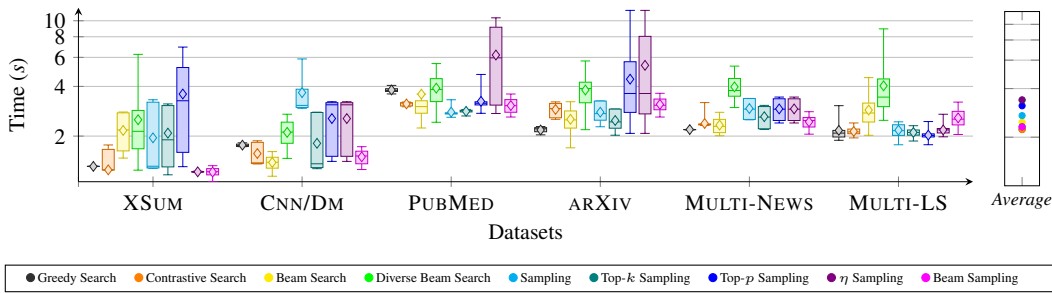

Figure 4: Decoding time complexity (in seconds) for summarizing a single instance of sampled dataset test sets. For each strategy, we also report the average time across all samples.

exceptionally well, always covering a podium position. When adequately tuned, Diverse Beam Search demonstrates its prowess independently of the AS type. Details on metric score distributions and per-dataset rankings are provided in Appendix H.

- **A2 Influenced by AS type, decoding strategies exhibit prevailing performance patterns across datasets.** Deterministic strategies show greater suitability for AS, with Sampling, Top-$k$ Sampling, and Top-$p$ Sampling struggling to keep up. The consistency of these patterns is evident when comparing similar benchmarks, such as PUBMED and ARXIV. A deviation is observed in MULTI-LEXSUM, which involves extremely long inputs (averaging >88K words, as shown in Appendix C). In this case, truncation narrows the gap between strategies, negatively affecting Contrastive Search. The hypothesized reason is the $\alpha>0$ hyperparameter, which assigns less weight to model confidence—a critical factor for handling uncertainty. Concerning AS types, Beam Search demonstrates robustness in MDS, Diverse Beam Search emerges as the preferred strategy in LDS, and Greedy Search proves particularly effective in SDS scenarios. These results raise warnings about claims made in previous publications with task-agnostic or single-AS-type settings (Su & Collier, 2023).

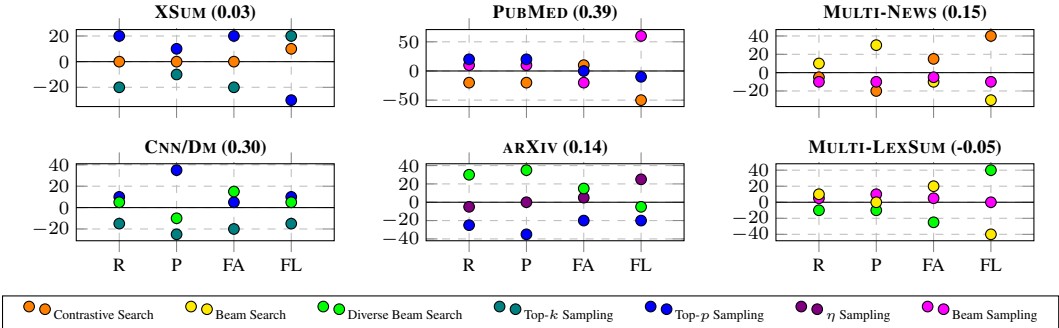

Figure 5: Win - Lose (%) human evaluation results on four quality dimensions: Recall (R), Precision (P), Faithfulness (FA), and Fluency (FL). The average Kendall's $\tau$ coefficients among all inter-annotator agreements are given in brackets.

- A3 **Changing the decoding strategy is not just about decimals.** The decoding heuristic can substantially affect the LM scores within a given benchmark dataset. This choice can result in variations of up to 20 $\mathcal{R}$ points in PUBMED, 9 BARTScore points in CNN/DM, and 278 Perplexity points in MULTI-LEXSUM.

- A4 **Deterministic methods generally balance effectiveness and efficiency.** Figure 4 reveals that Greedy Search, Beam Search, and Beam Sampling are the fastest strategies. For Diverse Beam Search, Top-$p$ Sampling, and $\eta$ Sampling, the latency is highly variable depending on the hyperparameters. Regarding $CO_2$ at inference time (cf. the green line in Figure 3 for Carburacy), Diverse Beam Search provides the best $\mathcal{R}$–efficiency trade-off. We refer the reader to Appendix H for Carburacy-ranked decoding strategies.

- A5 **Not all quality attributes are easy to temper at decoding time.** High beam size, high `no_repeat_ngram_size`, high temperature, and low diversity penalty promote factuality and semantic consistency. When transitioning to $\mathcal{R}$, it is strongly advised to maintain a large beam size, while the use of 0.8 temperatures in Top-$k$ Sampling and Top-$p$ Sampling frequently results in degeneration clusters. Interestingly, no strategy or hyperparameter can greatly favor the naturalness of text in a predictable way. However, we observe a strong positive correlation between Perplexity and Density scores (0.73 Pearson coeff.). We offer a thorough examination in Appendix I, looking at how metrics respond to fine-grained variations in hyperparameters. Appendix J elucidates the best hyperparameter values across all datasets, while Appendix K shows a per-dataset evaluation.

- **Redundancy is ubiquitous, mainly in MDS.** Our scores contradict Meister et al. (2022b), referring to redundancy as a rare phenomenon in AS. We pinpoint a tendency for recurring tokens as the input length increases, peaking with MULTI-LEXSUM (-60.97% UNR avg.).

- **Stochastic vs. deterministic.** Deliberate addition of randomness increases the chance of unconventional summaries and contradictions. Sampling, Top-$k$ Sampling, and Top-$p$ Sampling have a larger sample variance than all other strategies on five of six datasets. As expected, they have fewer repetitions (+27.63% UNR avg.) (Fan et al., 2018; Holtzman et al., 2020) but tend more to factual flaws (-40.58% BARTScore avg.) (Basu et al., 2021; Su et al., 2022). Notably, $\eta$ Sampling and Beam Sampling stand out in the stochastic sphere, suffering less from hallucinations (+40.23% BARTScore avg.). Together with deterministic methods, they produce the highest $\mathcal{R}$ (+29.65% avg.) and BERTScore (+35.34% avg.).

- **The output length matters.** Short summaries (i.e., those in XSUM and CNN/DM) are less redundant than longer ones (+46.59% UNR avg.) (Xiao & Carenini, 2020a).

## 5.2 QUALITATIVE FINDINGS

The annotation process took approximately 6 hours per judge. The results are presented in Figure 5. The average Kendall's $\tau$ of 0.16, calculated between two annotators across all pair selection results, reflects high competitiveness among the top-3 decoding strategies when correctly tuned, oftentimes leading to subjective summary preferences.

- **Top-$p$ Sampling shines in AS with concise inputs.** While not indicated by the $\mathcal{R}$ metric on XSUM, Top-$p$ Sampling gains SDS human preference by 75%. However, its effectiveness fades as the input length increases, tipping the balance in favor of deterministic alternatives. According to this principle, Beam Search prevails in MDS.

- **Fluency negatively correlates with Recall, Precision, and Factuality**. Fluency goes opposite to Recall, Precision, and Factuality more than 66% of the time. Furthermore, Fluency preferences depend on the dataset and do not always reward stochastic strategies.

## 6 THE PRISM DATASET

**Composition**   Building upon the experiments presented in the previous sections, we introduce PRISM, a first-of-its-kind dataset that collects over 2M artificial summaries generated over a range of heterogeneous decoding settings. PRISM presents an instance for each inference run, detailing all metadata (dataset, model, decoding strategy, hyperparameter values), average decoding time per instance (milliseconds), carbon emissions (kg), and metric scores. Its source files ($\approx$10 GB) are stored in JSONL format and

```python
from datasets import load_dataset
# Download PRISM locally and load it as a Dataset
prism = load_dataset("PRISM")
# The first Beam Search run
run = prism["beam_search"][0]
# The predicted summaries
run["predictions"]
```

Figure 6: PRISM HuggingFace Dataset.

are publicly available for download through the HuggingFace Datasets platform.[6] For example, to access the summaries predicted by a Beam Search run, you need to install the datasets Python library and follow the instructions shown in Figure 6. For the sake of space efficiency, we separately release the gold document-summary AS pairs that each run relies on. Additional information on the project website[7] will be updated regularly to incorporate any future changes, additions, or erratum.

**Applications**   The potential applications of PRISM are extensive and diverse. Researchers can exploit this dataset to study new NLG metrics (Frisoni et al., 2022a). Additionally, it provides a unique opportunity to benchmark decoding strategies against a multitude of established baselines. Beyond this, PRISM offers the ability to train LMs to emulate the token choice of one or more strategies for style control (Goyal et al., 2022) or automatic hyperparameter optimization (Chen et al., 2022). In fact, decoding strategies are complex algorithms that are hard to put in end-to-end networks due to their non-differentiable nature. In light of the poor attempts to design exact differentiable versions of decoding strategies (e.g., Top-$k$ Sampling (Jang et al., 2017), Beam Search (Collobert et al., 2019)), PRISM emerges as an indispensable asset for creating approximated modules.

## 7 CONCLUSION

The rocketing growth witnessed by transformer-based summarizers is offset by the poor control over decoding strategies, which exhibit cloudy task-specific qualities overshadowed by the continuous distribution of new models. In this paper, we demystify the role of decoding-time methods for abstractive summarization. Our full-scale study comprises comprehensive quantitative and qualitative breakdowns, covering various decoding setups, autoregressive models, datasets, and evaluation metrics. Empirical results demonstrate how generative heuristics and their hyperparameters can overturn predicted summaries, where optimal choices depend on target quality dimensions and the summarization type at hand (i.e., long, short, multi-document). Besides validating observations already made in other tasks, our findings unveil the uniqueness of abstractive summarization and the best procedures to follow depending on the case, serving as cautionary notes. Wrapping up our core findings, we furnish practitioners with a practical and easy-to-follow guideline (Appendix L), facilitating the right selection of decoding strategies and hyperparameters tailored to specific case studies. Our breath study and the data collected unlock new research avenues, raising expectations for a future marked by increased awareness of the implications of decoding and their control.

---

[6] https://huggingface.co/datasets/[anonymized]/PRISM
[7] https://prism.github.io. The web page will be public at the end of the anonymization window.

## REPRODUCIBILITY STATEMENT

To help readers reproduce our experiments, we provide rationales for our decoding hyperparameter search space in Appendix D, listing each dataset's minimum and maximum input/output length. All models, datasets, decoding strategies, and automatic metrics explored in this study are open source; Appendix E elaborates on implementation specifics, hardware setup, and runtimes. Since our decoding runs are performed on representative dataset samples, we also include details of the preprocessing steps and the power analysis process completed before sampling (Appendix C). Appendix F describes our human evaluation protocol. We plan to openly release the source codes in a dedicated GitHub repository and PRISM on HuggingFace Datasets.

## ETHICS STATEMENT

We honor ICLR Code of Ethics. As we recognize that the reported principles are not exhaustive, we address the nine points explicitly mentioned in the NeurIPS 2023 Ethical Guidelines.

**1. Does the data contain any personally identifiable information or sensitive personally identifiable information?** Our data do not contain confidential information. All source documents are available for free inspection, uncopyrighted, and fully public.

**2. Does the data contain information that could be deduced about individuals that they have not consented to share?** Our data contain individual names. Nevertheless, such details are within news reports published by authoritative sources, such as BBC and CNN, on which we rely.

**3. Does the data encode, contain, or potentially exacerbate bias against people of a certain gender, race, sexuality, or who have other protected characteristics?**
No.

**4. Does the paper contain human subject experimentation and whether it has been reviewed and approved by a relevant oversight board?**
No.

**5. Does the paper rely on data that have been discredited by the creators?**
No.

**6. Consent to use or share the data. Explain whether you have asked the data owner's permission to use or share data and what the outcome was.**
Consent is implicit for all content because the datasets used are publicly available.

**7. Domain specific considerations when working with high-risk groups**
Not applicable.

**8. Filtering of offensive content. For instance, when collecting a dataset, how are the authors filtering offensive content such as racist language or violent imagery?**
Not applicable. Our data does not involve offensive content.

**9. Compliance with GDPR and other data-related regulations. For instance, if the authors collect human-derived data, what is the mechanism to guarantee individuals' right to be forgotten (removed from the dataset)?**
As noted in the paper, our data are derived from multiple publicly available datasets, including news reports, scientific papers, and court lawsuits that do not comply with privacy rules.

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

## A ABSTRACTIVE SUMMARIZATION FAMILIES

We define the problem of AS with the following setup. For single-source tasks (SDS and LDS), the input is a document $\mathcal{X} = \{x_1, \ldots, x_x\}$, where each $x_i \in \mathcal{X}$ is a token. For multi-source scenarios (MDS), the input is a cluster $\mathcal{C} = \{\mathcal{X}_1, \ldots, \mathcal{X}_z\}$ of documents.

- In *SDS*, $\mathcal{X}$ is a brief document, generally shorter than 1024 tokens, which is the maximum size that transformer-based LMs with quadratic complexity (Lewis et al., 2020; Zhang et al., 2020a) can process without input truncation.
- In *LDS*, the number of input tokens could be potentially large (e.g., > 10,000). For this reason, quadratic LMs would ignore summary-worthy information and are thus replaced by efficient transformers with linear complexity that can read up to 16,384 tokens (Beltagy et al., 2020; Huang et al., 2021b; Guo et al., 2022; Phang et al., 2022).
- In *MDS*, $\mathcal{C}$ is a cluster consisting of multiple documents related to a topic (e.g., newspaper articles detailing the same recent event). Generally, $\mathcal{X}$ is assembled by concatenating the documents in $\mathcal{C}$ to form a single long textual input (DeYoung et al., 2021; Xiao et al., 2022), treating the summarization problem as in LDS.

## B SCOPUS QUERIES

The bibliometric results reported in the main paper (Figure 2) are obtained by executing the following queries on the SCOPUS search engine:

- **Abstractive Summarization**
  ```
  TITLE-ABS-KEY(abstractive AND summarization);
  ```
- **Decoding Strategies**
  ```
  TITLE-ABS-KEY(((text AND generation) OR nlg) AND decoding);
  ```
- **Abstractive Summarization + Decoding Strategies**
  ```
  TITLE-ABS-KEY(summarization AND decoding);
  ```

We also show in Figure 7 the intersection between abstractive summarization and large language models (LLMs), obtained with the subsequent query:

- **Abstractive Summarization + Large Language Models**
  ```
  TITLE-ABS-KEY(abstractive AND summarization AND ((large AND
  language AND (model OR models)) OR LLM)).
  ```

We only consider conference papers on Computational Science by appending the following: `AND (LIMIT-TO(SUBJAREA, "COMP")) AND (LIMIT-TO(DOCTYPE, "cp"))`.

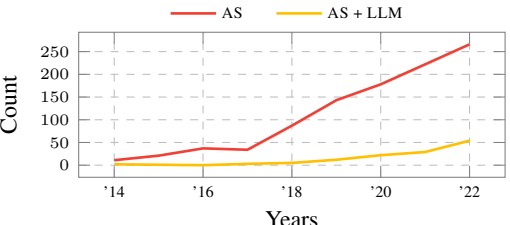

Figure 7: Annual rate SCOPUS comparison between conference papers on abstractive summarization (AS) and their intersection with large language models (LLMs).

## C DATASET PREPROCESSING AND POWER ANALYSIS

According to best practices in summarization corpora construction (Kornilova & Eidelman, 2019b), our preprocessing pipeline foresees (i) lowercasing, (ii) ASCII encoding, (iii) extra space, URL,

special character, and bullet point removal, (iv) HTML cleaning, (v) dash sequences deletion, (vi) newline and tabs erasure, (vii) empty sentence and non-alphabetic starter cut, (viii) between-sentence demarcation space assurance, (ix) word contraction expansion, and (x) quotes normalization.

A prudent preliminary operation is to determine the required sample size to ensure that the study has sufficient statistical power to detect meaningful metric effects across decoding runs if they exist. To this end, we perform a power analysis for the t-tests using the `statsmodels` Python library (v 0.14.0). The resulting needed sample size is 393.4. Technically, we consider the following factors:

- *Power level*. It denotes the probability of correctly detecting a true effect if it exists. Higher power indicates greater ability to detect effects. We use the commonly recommended value 0.8, which corresponds to an 80% chance of correctly detecting a true effect.

- *Significance level*. It is the threshold used to determine statistical significance. We choose 0.05, which corresponds to a 5% chance of incorrectly rejecting the null hypothesis.

- *Effect size*. It is the magnitude of the difference that we expect to observe among different decoding strategies. We employ Cohen's *d*, calculated by taking the difference in means between two groups and dividing it by the pooled standard deviation, i.e., $(\bar{x_1} - \bar{x_2})/s$. Following preliminary experiments, we make an educated guess about the aggregate effect size across all NLG metrics. We set $\bar{x_1} - \bar{x_2}$ at 0.01 and $s$ at 0.05.

After preprocessing, the dataset statistics (before and after sampling) are listed in Table 4. The number of words is derived from `nltk.word_tokenize` (Bird, 2006).

Table 4: Dataset statistics before and after sampling, including test set size, number of sources per instance, and the words in the source and target texts. All values are averaged except "# Samples."

| | Before sampling | | Source | | Target | After sampling | | Source | | Target |
|---|---|---|---|---|---|---|---|---|---|---|
| | Dataset | # Samples | # Docs | # Words | # Words | Dataset | # Samples | # Docs | # Words | # Words |
| *SDS* | XSUM | 11,333 | 1 | 421.2 | 22.9 | XSUM | 1134 | 1 | 414.1 | 22.8 |
| | CNN/DM | 11,490 | 1 | 761.3 | 57.4 | CNN/DM | 1148 | 1 | 753.8 | 58.0 |
| *LDS* | PUBMED | 6658 | 1 | 3070.4 | 206.7 | PUBMED | 667 | 1 | 3095.5 | 207.2 |
| | ARXIV | 6440 | 1 | 5733.2 | 161.6 | ARXIV | 644 | 1 | 5770.7 | 160.6 |
| *MDS* | MULTI-NEWS | 5621 | 2.8 | 2026.3 | 245.9 | MULTI-NEWS | 555 | 2.8 | 1947.9 | 248.0 |
| | MULTI-LEXSUM | 616 | 10.3 | 88,122.9 | 126.5 | MULTI-LEXSUM | 616 | 10.3 | 88,122.9 | 126.5 |

## D  DECODING HYPERPARAMETERS

For *Contrastive Search*, we modulate $\alpha$ and $k$ in $\{0.2, 0.4, 0.6, 0.8, 1.0\}$ and $\{20, 30, 40, 50, 60\}$, respectively. For *Beam Search*, we choose the beam size $b$ from $[2, 10]$, a large window that allows a trade-off between quality and performance. We keep $b$ unchanged for *Diverse Beam Search*, implementing $\Delta$ with the Hamming distance, as suggested by Vijayakumar et al. (2018), and considering a diversity penalty $\lambda \in \{0.2, 0.4, 0.6, 0.8, 1.0\}$; the number of subgroups is set equal to $b$. We keep the same exploration range for $b$ when we try out *Beam Sampling*. For *Top-k Sampling*, we pick $k$ from $\{20, 30, 40, 50, 60\}$. For *Top-p Sampling*, supported by the results of DeLucia et al. (2021), we choose $p$ from $\{0.4, 0.6, 0.8\}$. For $\eta$ *Sampling*, we search $\eta$-cutoff over $\{0.0003, 0.0006, 0.0009, 0.002, 0.004\}$ (Hewitt et al., 2022). For pure (ancestral), Top-$k$, and Top-$p$ Sampling, we also perform an ablation on the temperature hyperparameter, exploring $\tau \in \{0.8, 0.9, 1.0\}$, the best values according to Holtzman et al. (2020) and Pasunuru et al. (2021). Where not specified, we maintain default hyperparameters depending on the model configuration.

Artificial summaries often suffer from high redundancy (Xiao & Carenini, 2020b). Therefore, we investigate non-negligible word-level $n$-grams penalties as introduced by Klein et al. (2017) and Paulus et al. (2018), precisely for AS. In a nutshell, we force the decoder to never output the same sequence of words more than once during testing by manually setting the probability of already hypothesized $n$-grams to 0. We extract $n$ from $\{3, 4, 5\}$ for PUBMED, ARXIV, MULTI-NEWS, and MULTI-LEXSUM, and from $\{2, 3\}$ for XSUM (one-line output) and CNN/DM. This setting gives a minimum of 16 runs per decoding strategy.

Table 5: List of the utilized datasets and relative length constraints.

| Dataset | URL | Input | Output | |
|---|---|---|---|---|
| | | Max Len | Min Len | Max Len |
| XSUM | https://huggingface.co/datasets/xsum | 1024 | 20 | 100 |
| CNN/DM | https://huggingface.co/datasets/cnn_dailymail | 1024 | 20 | 100 |
| PUBMED | https://huggingface.co/datasets/ccdv/pubmed-summarization | 4096 | 100 | 256 |
| ARXIV | https://huggingface.co/datasets/ccdv/arxiv-summarization | 4096 | 100 | 256 |
| MULTI-NEWS | https://huggingface.co/datasets/multi_news | 4096 | 100 | 256 |
| MULTI-LEXSUM | https://huggingface.co/datasets/allenai/multi_lexsum | 4096 | 50 | 256 |

Table 6: Related work on decoding strategies comparison.

| Source | Decoding Strategies | Hyperparameters[*] | Task[**] | Automatic Metrics | Systematic Benchmark |
|---|---|---|---|---|---|
| Holtzman et al. (2020) | Greedy Search
Beam Search
Sampling
Top-$k$ Sampling
Top-$p$ Sampling | /
$b \in \{4, 8, 16\}$
$\tau \in [0.1, 0.2, \dots, 1.0]$
$k \in 5 \times 2^{[1,2,\dots,12]}$
$\approx p \in [0.1, 0.2, \dots, 0.9, 0.95]$ | OEG | Perplexity, Self-BLEU, Repetition, Zipf Coefficient | ✗ |
| Meister et al. (2022b) | Greedy Search
Beam Search
Diverse Beam Search
Sampling
Top-$k$ Sampling
Top-$p$ Sampling | /
$b \in \{5, 10\}$
$\Delta = $ Hamming, $\lambda = 0.7, |g| = b = 5$
/
$k = 30$
$p = 0.85$ | OEG,
MT,
SDS,
DG,
SG | BLEU, COMET, ROUGE BLEURT, DIST-$n$, ENT-$n$, $n$-GRAM DIV., Self-BLEU, Repetition | ✔ |
| Ippolito et al. (2019) | Beam Search
Diverse Beam Search
NPAD Beam Search
Clustered Beam Search
Top-$g$ Capping Beam Search
Iterative Beam Search
Sampling | $b = 10$
$\Delta = $ Hamming, $\lambda = 0.8, |g| = b = 10$
$b = 10, \sigma_0 = 0.3$
$b = 10, c = 5$
$b = 10, g\_c = 3$
$b = 10, i = 5$
$\tau \in \{0.5, 0.7, 1.0\}$ | DG,
IC | Perplexity, DIST-$n$, ENT-$n$, SPICE | ✔ |
| Leblond et al. (2021) | Greedy
Beam Search
Value-Guided Beam Search
Monte Carlo Tree Search
Sampling + Ranking Variants | /
$b \in [2, 4, \dots, 10, 20]$, logits $\tau \in [0.6, 0.8, \dots, 1.4]$
normalization $\tau \in [0.4, 0.6, \dots, 1.0]$
$b = 6$, logits $\tau \in [0.6, 0.8, \dots, 1.4]$,
$\alpha_{lc} \in [0, 0.1, \dots, 0.9, 0.95, 1.0]$
logits $\tau \in \{0.9, 1.1, 1.3\}, c_{puct} \in [1.0, 2.0, \dots, 6.0, 8.0]$
$\tau \in [0.15, 0.25, \dots, 0.95]$ | MT | BLEU, BERTScore | ✗ |
| Su et al. (2022)
Su & Collier (2023) | Greedy
Contrastive Search
Beam Search
Typical Sampling
Top-$k$ Sampling
Top-$p$ Sampling | /
$k \in [2, 3, \dots, 10], \alpha \in [0.1, 0.2, \dots, 1]$
$b \in \{4, 5, 10\}$
$\tau = 0.95$
$k = 50$
$p = 0.95$ | OEG,
DG,
SDS,
CG,
MT | Perplexity, Accuracy, Repetition, $n$-GRAM DIV., MAUVE, Semantic Coherence | ✗ |
| Ours | Greedy
Contrastive Search
Beam Search
Diverse Beam Search
Sampling
Top-$k$ Sampling
Top-$p$ Sampling
Beam Sampling
$\eta$ Sampling | no_repetition_ngram_size $\in \{2, 3, 4, 5\}$
no_repetition_ngram_size $\in \{2, 3, 4, 5\}$,
$k \in [20, 30, \dots, 60], \alpha \in [0.2, 0.4, \dots, 1]$
no_repetition_ngram_size $\in \{2, 3, 4, 5\}$,
$b \in [2, 3, \dots, 10]$
no_repetition_ngram_size $\in \{2, 3, 4, 5\}$,
$\Delta = $ Hamming, $\lambda \in [0.2, 0.4, \dots, 1.0]$,
$|g| = b \in [2, 3, \dots, 10]$
no_repetition_ngram_size $\in \{2, 3, 4, 5\}$,
$\tau \in \{0.8, 0.9, 1.0\}$
no_repetition_ngram_size $\in \{2, 3, 4, 5\}$,
$\tau \in \{0.8, 0.9, 1.0\}, k \in [20, 30, \dots, 60]$
no_repetition_ngram_size $\in \{2, 3, 4, 5\}$,
$\tau \in \{0.8, 0.9, 1.0\}, k \in [\text{None}, 20, 30, \dots, 60]$
$p \in 0.4, 0.6, 0.8$
no_repetition_ngram_size $\in \{2, 3, 4, 5\}$,
$b \in [2, 3, \dots, 10], p = 0.9$
no_repetition_ngram_size $\in \{2, 3, 4, 5\}$,
$\eta \in \{0.0003, 0.0006, 0.0009, 0.002, 0.004\}$ | SDS,
LDS,
MDS | ROUGE, BERTScore, Perplexity, Coverage, Density, Compression, UNR, NID, BARTScore-F, Carburacy + Runtime | ✔ |

[*] $b = $ beam size, $\tau = $ temperature, $k/p = $ top-$k/p$ thresholds for selecting candidate pools, $\Delta = $ diversity measure, $\lambda = $ diversity penalty, $|g| = $ number of sub-groups, $\sigma_0 = $ decoder-level random noise, $c = $ number of clusters, $g\_c = $ top parent hypotheses considered at each time step, $i = $ number of iterations, $\alpha_{lc} = $ linear combination weights, $\eta = $ entropy-dependent cutting-off.

[**] OEG = Open-Ended (Unconditional) Generation, MT = Machine Translation, DG = Dialogue Generation, SG = Story Generation, IC = Image Captioning, CG = Code Generation, SDS = Short Document Summarization, LDS = Long Document Summarization, MDS = Multi-Document Summarization.

As common in AS, we define a minimum and maximum summary length for each dataset (Table 5), avoiding the production of an EOS token before or after a certain threshold. These constraints are determined by examining the summary sizes in the validation tests.

**Decoding Runs** There is a strong paucity of research on generative effects governed by the available next-token selection strategies. To our knowledge, only two systematic benchmarks focus on intertask behaviors and output diversity (Ippolito et al., 2019; Meister et al., 2022b). Most comparative studies originate primarily from papers that introduce novel decoding techniques with the intention of providing either theoretical justification or empirical evidence for their effectiveness. Although the selection of the decoding strategy is widely known to be task-dependent, the emphasis in community efforts primarily lies on open-ended generation, leaving several questions unanswered in application fields such as AS. In this area, Beam Search is often lazily treated as the de-facto standard, with limited exploration of alternative decoding strategies (Beltagy et al., 2020; Lewis et al., 2020; Xiao

Table 7: List of the utilized models.

| Model | Dataset | URL |
|-------|---------|-----|
| BART-large | XSUM | https://huggingface.co/facebook/bart-large-xsum |
| | CNN/DM | https://huggingface.co/facebook/bart-large-cnn |
| LED-large | PUBMED | https://huggingface.co/patrickvonplaten/led-large-16384-pubmed |
| | ARXIV | https://huggingface.co/allenai/led-large-16384-arxiv |
| PRIMERA-large | MULTI-NEWS | https://huggingface.co/allenai/PRIMERA-multinews |
| | MULTI-LEXSUM | https://huggingface.co/allenai/primera-multi_lexsum-source-short |

et al., 2022). Despite the collective superficiality and the latest heuristics, we want to step back and question this leap of faith. Furthermore, research efforts that already incorporate AS often draw task-level conclusions based on the analysis of a single dataset and a limited research scope, resorting to SDS for the sake of experimental efficiency (Meister et al., 2022b; Su & Collier, 2023). Fanning the flames, the efficiency data on runtime and $CO_2$ are rarely mentioned. Compared to previous work (Table 6), our task-specific investigation is offset by the high diversity of heuristics and their settings, detailing the characteristics of the SDS/LDS/MDS family. Consequently, we consider this work the most extensive decoding study, also w.r.t. the diversity of AS models evaluated. For example, previous research by Meister et al. (2022b) and Su et al. (2022); Su & Collier (2023) considered exclusively one AS model each: BART (Lewis et al., 2020) and GPT-2 (Radford et al., 2019), respectively.

# E IMPLEMENTATION DETAILS

**Hardware Configuration**  Inference runs are performed on a workstation that has 4 Nvidia GeForce RTX3090 GPUs with 24 GB of dedicated memory each, 64 GB VRAM, and an Intel® Core™ i9-10900X1080 CPU @ 3.70GHz.

**Models, Datasets, and Decoding**  Our code is founded on PyTorch 1.10.2 (Paszke et al., 2019), the HuggingFace Transformers (Wolf et al., 2020) and Datasets (Lhoest et al., 2021) libraries. Table 5 and Table 7 enumerate datastore references and model checkpoints fine-tuned on them. Sticking to Xiao et al. (2022), the input for PRIMERA is the concatenation of documents within the clusters (in the same order), where truncation is applied to each document based on the input length limit divided by the size of the cluster, thus guaranteeing the representation of all sources. All decoding runs for autoregressive summary generation are subject to the generate() method of HuggingFace. We set the global seed to 42 to guarantee the reproducibility of all runs of our work.

**Metrics**  We quantify automatic metric scores using NLG-METRICVERSE (Frisoni et al., 2022a), moving to external official repositories where it is impossible. Table 8 lists all hyperparameters. Due to the greater correlation with human judgment, we computed BERTScore with DEBERTA-large instead of the default ROBERTA-large, as recommended by the authors from version 0.3.11. We conducted tests using SummaC (Laban et al., 2022), which is considered the main factuality metric for abstractive LDS (Koh et al., 2022). However, its computational demands proved to be too time-consuming, leading us to prefer BARTScore.

Table 8: Hyperparameters of the NLG metrics.

| Metric | Hyperparameters |
|--------|-----------------|
| ROUGE | rouge_types=["rouge1","rouge2","rougeLsum"], use_stemmer=True |
| BERTScore | lang="en", model_type="microsoft/deberta-large-mnli", idf=True, batch_size=32, rescale_with_baseline=True |
| Perplexity | model_id="gpt2", add_start_token=False |
| BARTScore | checkpoint="facebook/bart-large-cnn", batch_size=4 |
| Carburacy | alpha=10, beta=100 |

**Experiment Tracking**  We tracked all our runs with Weights & Biases[8] and monitor $CO_2$ emissions with CodeCarbon.[9] Our experiments can be publicly found on our Weights and Biases project.[10] The cumulative processing time of the GPU for

---

[8] https://wandb.ai

[9] https://github.com/mlco2/codecarbon

[10] https://wandb.ai/decoding-summ/prism

Table 9: Average FLOPs computational performance of the evaluated decoding strategies.

| Greedy | Contrastive | Beam Search | DBS | Sampling | Top-$k$ | Top-$p$ | $\eta$ Sampling | Beam Sampling |
|--------|-------------|-------------|-----|----------|---------|---------|-----------------|---------------|
| 1.7e17 | 1.7e17 | 1.9e17 | 2.6e17 | 2.1e17 | 1.8e17 | 2.4e17 | 2.6e17 | 1.8e17 |

generating all summaries amounts to $\approx$60 days. Computing all metric scores requires $\approx$13 days: 12 hours for $\mathcal{R}$, 18 hours for BERTScore, 3.6 hours for Perplexity, 198 hours for Coverage, Density, and Compression, 38 minutes for UNR and NID, 75 hours for BARTScore, and 10 seconds for Carburacy. Table 9 reports FLOPs computational performance of the decoding strategies as an additional measure of processing power and efficiency.

## F  HUMAN EVALUATION

Our setup is sketched in Figure 8, inspired by Gu et al. (2022). We instruct the annotators to distinguish quality dimensions by presenting the example reported in Table 10.

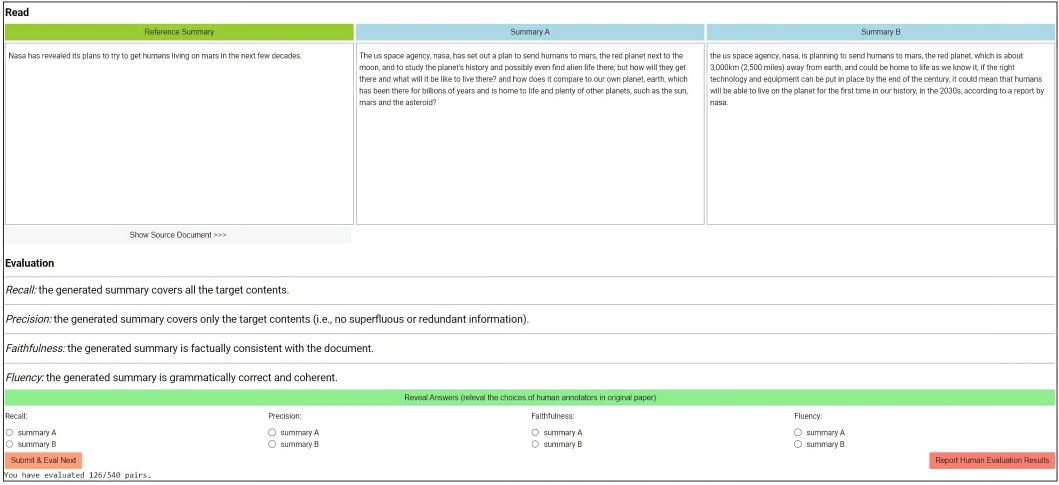

Figure 8: Screenshot of the interface used to perform the human evaluation.

Table 10: Example attached to annotation instructions.

| Document |
|---|
| African elephants are the largest land animals on Earth. They are known for their distinctive large ears and tusks, which are actually elongated incisor teeth. These gentle giants live in herds led by a matriarch and have a complex social structure. African elephants face threats from poaching and habitat loss, which have led to a decline in their population. |
| **Gold Summary** |
| African elephants live in herds led by a matriarch. Despite their gentle nature, they face endangerment due to poaching and habitat loss, leading to a declining population. **Topics**: African elephants - Social structure - Threats they face. |
| **Summary 1 (High Recall, High Precision, Low Faithfulness)** |
| African elephants live in herds led by a matriarch. They are not endangered and face no threats. **Explanation**: Summary 1 has a *high Recall* because it includes all relevant information/topics of the gold summary; it has a *high Precision* because it includes only the relevant information/topics without adding irrelavant details; the *Faithfulness is low* because it includes incorrect information about their endangerment status. |
| **Summary 2 (High Recall, Low Precision, Low Faithfulness)** |
| African elephants, the Earth's largest land animals, are known for their large ears and tusks. They live individually and face threats from poaching and habitat loss, causing a decline in their population. **Explanation**: Summary 2 has a *high Recall* because it includes all relevant information/topics of the gold summary; it has a *low Precision* because it also includes additional information/topics that are not within the gold summary; the *Faithfulness is low* because it includes incorrect information about their social structure. |

## G    NORMALIZED TABULATED SCORES

For better interpretability, Table 11 offers the tabulated version of Figure 3. Data are categorized by dataset, metric, and decoding strategy, with a 0-10 normalization imposed by the magnitude of values and graphical reasons. We prioritize the relative relationships among the strategies rather than the precise score values achieved by each. Please note that the exact metric scores are openly released in our PRISM dataset.

Table 11: Normalized tabulated scores categorized by dataset, metric, and decoding strategy. For each dataset, the decoding strategies are sorted by decreasing AVG.

| Strategy | $\mathcal{R}$ | BERTScore | Perplexity↓ | Coverage | Density↓ | Compression | UNR | NID↓ | BARTScore | Carburacy |
|---|---|---|---|---|---|---|---|---|---|---|
| **XSUM** | | | | | | | | | | |
| Beam Sampling | 3.44 | 7.31 | 1.49 | 5.95 | 0.31 | 0.03 | 6.67 | 4.65 | 7.51 | 4.25 |
| Greedy Search | 3.25 | 8.05 | 1.28 | 6.28 | 0.35 | 0.04 | 5.96 | 6.22 | 8.11 | 4.24 |
| Contrastive Search | 3.27 | 8.01 | 1.37 | 6.17 | 0.32 | 0.04 | 5.97 | 6.16 | 8.09 | 4.27 |
| $\eta$ Sampling | 3.27 | 8.01 | 1.40 | 6.08 | 0.35 | 0.03 | 5.96 | 6.17 | 8.10 | 4.26 |
| Beam Search | 3.30 | 7.97 | 1.30 | 6.27 | 0.35 | 0.04 | 5.85 | 6.37 | 8.06 | 4.28 |
| Diverse Beam Search | 3.80 | 6.82 | 1.65 | 5.96 | 0.28 | 0.05 | 4.48 | 8.21 | 7.25 | 4.80 |
| Sampling | 2.34 | 5.16 | 1.37 | 3.78 | 0.14 | 0.03 | 7.51 | 3.54 | 5.21 | 3.04 |
| Top-$k$ Sampling | 2.22 | 5.48 | 1.37 | 4.23 | 0.23 | 0.03 | 7.09 | 4.25 | 5.45 | 2.90 |
| Top-$p$ Sampling | 2.30 | 5.34 | 1.48 | 4.37 | 0.25 | 0.03 | 6.88 | 4.37 | 5.41 | 3.00 |
| **CNN/DM** | | | | | | | | | | |
| Greedy Search | 8.28 | 9.88 | 6.55 | 9.94 | 5.22 | 0.07 | 7.33 | 4.40 | 9.91 | 8.77 |
| $\eta$ Sampling | 8.25 | 9.86 | 6.24 | 9.70 | 5.26 | 0.07 | 7.32 | 4.40 | 9.89 | 8.76 |
| Beam Search | 8.30 | 9.87 | 6.49 | 9.98 | 5.58 | 0.08 | 7.42 | 4.40 | 9.93 | 8.80 |
| Diverse Beam Search | 8.44 | 9.90 | 6.12 | 9.86 | 6.12 | 0.08 | 7.30 | 4.49 | 9.80 | 8.92 |
| Beam Sampling | 8.20 | 9.79 | 6.36 | 9.71 | 5.28 | 0.08 | 7.36 | 4.38 | 9.84 | 8.48 |
| Contrastive Search | 8.26 | 9.87 | 6.27 | 9.93 | 6.20 | 0.09 | 7.31 | 4.40 | 9.89 | 8.77 |
| Top-$k$ Sampling | 6.51 | 7.17 | 6.24 | 6.49 | 3.01 | 0.02 | 8.23 | 2.95 | 6.82 | 7.14 |
| Sampling | 6.46 | 7.12 | 6.30 | 6.49 | 3.08 | 0.02 | 8.20 | 2.94 | 6.79 | 6.99 |
| Top-$p$ Sampling | 5.67 | 5.53 | 6.15 | 6.78 | 3.07 | 0.08 | 8.45 | 2.27 | 5.02 | 6.37 |
| **PUBMED** | | | | | | | | | | |
| Diverse Beam Search | 8.75 | 8.81 | 3.95 | 9.73 | 7.01 | 0.23 | 3.43 | 5.71 | 8.44 | 9.10 |
| Contrastive Search | 9.14 | 9.07 | 4.22 | 9.87 | 8.55 | 0.20 | 3.70 | 5.40 | 8.59 | 9.41 |
| Greedy Search | 9.11 | 9.06 | 4.25 | 9.86 | 8.89 | 0.19 | 3.82 | 5.34 | 8.65 | 9.31 |
| Beam Search | 8.97 | 8.99 | 4.17 | 9.86 | 9.13 | 0.19 | 3.98 | 5.28 | 8.71 | 9.26 |
| Beam Sampling | 9.04 | 9.03 | 4.23 | 9.73 | 8.88 | 0.19 | 3.85 | 5.34 | 8.62 | 9.24 |
| $\eta$ Sampling | 9.13 | 9.07 | 4.23 | 9.75 | 8.90 | 0.18 | 3.71 | 5.39 | 8.59 | 9.27 |
| Top-$k$ Sampling | 5.71 | 5.33 | 4.24 | 6.35 | 4.75 | 0.22 | 6.66 | 2.75 | 4.58 | 6.32 |
| Sampling | 5.74 | 5.33 | 4.32 | 6.34 | 4.71 | 0.21 | 6.64 | 2.75 | 4.57 | 6.33 |
| Top-$p$ Sampling | 5.81 | 5.32 | 3.99 | 6.33 | 4.69 | 0.24 | 6.31 | 2.99 | 4.55 | 6.36 |
| **ARXIV** | | | | | | | | | | |
| Diverse Beam Search | 9.16 | 8.37 | 8.58 | 9.51 | 3.66 | 0.81 | 3.84 | 6.34 | 7.54 | 9.35 |
| $\eta$ Sampling | 9.53 | 8.64 | 9.12 | 9.73 | 5.65 | 0.66 | 3.91 | 6.06 | 7.71 | 9.59 |
| Contrastive Search | 9.50 | 8.64 | 9.30 | 9.72 | 5.83 | 0.67 | 3.92 | 6.06 | 7.72 | 9.77 |
| Beam Sampling | 9.47 | 8.62 | 9.11 | 9.73 | 5.68 | 0.64 | 3.94 | 6.04 | 7.74 | 9.30 |
| Greedy Search | 9.45 | 8.59 | 9.32 | 9.73 | 6.62 | 0.65 | 3.86 | 6.02 | 7.85 | 9.62 |
| Beam Search | 9.37 | 8.57 | 9.29 | 9.79 | 7.14 | 0.63 | 3.90 | 5.97 | 7.91 | 9.55 |
| Top-$p$ Sampling | 7.10 | 5.97 | 9.29 | 6.62 | 3.39 | 0.64 | 5.84 | 3.95 | 4.68 | 7.62 |
| Sampling | 7.02 | 6.05 | 9.40 | 6.64 | 3.82 | 0.62 | 6.11 | 3.78 | 4.76 | 7.56 |
| Top-$k$ Sampling | 7.04 | 6.06 | 9.48 | 6.66 | 3.88 | 0.62 | 6.11 | 3.79 | 4.78 | 7.55 |
| **MULTI-NEWS** | | | | | | | | | | |
| Greedy Search | 8.69 | 8.93 | 0.33 | 8.38 | 1.32 | 0.15 | 4.39 | 5.81 | 8.36 | 9.05 |
| Beam Search | 8.64 | 8.90 | 0.35 | 8.35 | 1.34 | 0.15 | 4.37 | 5.82 | 8.35 | 9.00 |
| Contrastive Search | 8.58 | 8.87 | 0.37 | 8.27 | 1.25 | 0.15 | 4.36 | 5.84 | 8.33 | 8.98 |
| $\eta$ Sampling | 8.60 | 8.88 | 0.36 | 8.18 | 1.25 | 0.15 | 4.38 | 5.82 | 8.32 | 8.95 |
| Beam Sampling | 8.49 | 8.79 | 0.38 | 8.14 | 1.24 | 0.14 | 4.31 | 5.85 | 8.27 | 8.90 |
| Diverse Beam Search | 8.16 | 8.65 | 0.43 | 8.09 | 0.88 | 0.16 | 4.07 | 6.15 | 8.18 | 8.61 |
| Top-$k$ Sampling | 6.13 | 5.69 | 0.37 | 5.36 | 0.75 | 0.14 | 6.54 | 3.50 | 4.90 | 6.77 |
| Sampling | 6.07 | 5.68 | 0.36 | 5.36 | 0.71 | 0.14 | 6.53 | 3.51 | 4.90 | 6.74 |
| Top-$p$ Sampling | 6.14 | 5.55 | 0.36 | 5.40 | 0.74 | 0.14 | 6.36 | 3.60 | 4.81 | 6.80 |
| **MULTI-LEXSUM** | | | | | | | | | | |
| Beam Search | 7.66 | 7.99 | 0.18 | 9.40 | 0.42 | 8.75 | 3.33 | 5.93 | 8.10 | 8.25 |
| Diverse Beam Search | 7.32 | 7.96 | 0.12 | 7.52 | 0.43 | 9.19 | 3.58 | 6.15 | 7.85 | 7.98 |
| Beam Sampling | 7.59 | 7.97 | 0.08 | 7.70 | 0.40 | 8.45 | 3.20 | 5.93 | 8.07 | 8.05 |
| Greedy Search | 7.09 | 7.59 | 0.10 | 8.61 | 0.40 | 8.70 | 1.82 | 6.69 | 7.83 | 7.81 |
| Contrastive Search | 5.89 | 6.87 | 0.52 | 8.59 | 0.44 | 8.81 | 3.42 | 5.75 | 7.26 | 6.79 |
| $\eta$ Sampling | 5.94 | 6.92 | 0.47 | 7.65 | 0.41 | 8.47 | 3.36 | 5.78 | 7.29 | 6.81 |
| Top-$p$ Sampling | 6.86 | 7.47 | 0.26 | 5.92 | 0.23 | 8.46 | 2.15 | 6.51 | 7.73 | 7.61 |
| Sampling | 6.13 | 6.99 | 0.37 | 5.78 | 0.23 | 8.66 | 3.10 | 5.93 | 7.38 | 6.98 |
| Top-$k$ Sampling | 6.12 | 7.02 | 0.42 | 5.75 | 0.22 | 8.33 | 3.16 | 5.93 | 7.38 | 6.98 |

## H    METRIC SCORE DISTRIBUTIONS

We show the distributions of the metric scores by decoding strategy and dataset in Figure 9. Depending on the hyperparameters chosen, Sampling, Top-$k$ Sampling, and Top-$p$ Sampling display the highest

degree of result variability. Out of the deterministic methods, Diverse Beam Search is characterized by high redundancy. MDS consistently yields a lower perplexity than the other task families, reflecting the difficulty of current models in achieving high naturalness. The extractive nature of a summary in LDS scenarios can vary greatly depending on the strategy implemented. As expected, the length of the target is strongly correlated with the $\mathcal{R}$ score, which implies a higher probability of generating n-grams of overlap with longer references, especially in LDS scenarios.

We offer per-dataset rankings on $\mathcal{R}$ (lexical overlap, Figure 10), BARTScore (semantic overlap and faithfulness, Figure 12), and Carburacy ($\mathcal{R}$–efficiency trade-off, Figure 11), recognized as useful metrics for practical applications. Visualizing the decoding strategy ranks offers a complementary perspective for a comparative fine-grained evaluation alongside individual and averaged observations. We found `temperature=0.8` in Top-$k$ Sampling and Top-$p$ Sampling being the reason for the pronounced negative metric score clusters in XSUM, CNN/DM, PUBMED, ARXIV, and MULTI-NEWS. The lowest ranks in MULTI-LEXSUM are attributable to $\eta$ Sampling and Contrastive Search.

## I  ON HYPERPARAMETERS EFFECT

To encourage a conscious configuration of decoding strategies, we now consider the relation between each hyperparameter value and the primary metrics (i.e., $\mathcal{R}$, BERTScore, BARTScore, Perplexity).

**Overall**  Unexpectedly, while the value of `no_repeat_ngram_size` (abbreviated as `no_rep`) rises, there is a decrease in $\mathcal{R}$ and BERTScore, but a simultaneous increase in factuality (BARTScore). Our conjecture is that this can be explained by the elevated abstractiveness of the datasets, a hypothesis reinforced by human evaluation. In fact, there is a tension between staying close to the source document and allowing abstractive modification. Taking a wider perspective, we find that `no_rep`, a frequently overlooked parameter, plays a substantial role in shaping the output.

**Greedy Search (Table 15)**  In the case of SDS and LDS, an increase in `no_rep` is accompanied by a corresponding increase in the value of $\mathcal{R}$, except for XSUM. MDS follows an inverse pattern: with the growth of `no_rep`, both $\mathcal{R}$ and BERTScore decline, while BARTScore increases. The data show a notable correlation between tasks, with only two outliers: $\mathcal{R}$ and BERTScore in XSUM, and Perplexity in MULTI-LEXSUM.

**Contrastive Search (Table 16)**  It is difficult to pinpoint the selection guidelines for `top_k`, but lower values achieve better results. Decreasing `no_rep` in XSUM leads to improved $\mathcal{R}$ scores. In MULTI-LEXSUM, choosing a low `penalty_alpha` is preferable.

**Beam Search (Table 17)**  When delving into the SDS scenarios, it becomes evident that factuality experiences enhancement as both `no_rep` and beam size $b$ increase. Looking at BERTScore, we also detect a positive trend with `no_rep` and $b$ as well, peaking at $b = 4$ and $b = 5$. On the contrary, when examining XSUM, there is a declining trend in $\mathcal{R}$ relative to high values for `no_rep` and $b$. However, this trend is less pronounced in CNN/DM, which benefits from a central $b$ and `no_rep` $= 3$. This dissimilarity can likely be ascribed to the varying length of the summaries. Specifically, since XSUM is much more concise, imposing a stringent non-repetition constraint can significantly affect lexical similarity. This opposition is also apparent in terms of Perplexity. As for LDS, ARXIV and PUBMED experience an increase in $\mathcal{R}$ and BERTScore with high `no_rep` and low $b$. Again, the value of $b$ sets a boundary on the attainable level of factuality. In the context of MDS, there is a remarkable shift in the trajectory of $\mathcal{R}$, which increases with `no_rep` and $b$. BARTScore maintains the same pattern as shown in SDS and LDS. BERTScore fluctuates more and confirms its dataset-specific conduct, but generally benefits from higher values of $b$, irrespective of `no_rep`.

**Diverse Beam Search (Table 18)**  In SDS, artificial summaries from XSUM achieve high $\mathcal{R}$ scores when there is an increase in `no_rep`, a decrease in $b$, and a low `diversity_penalty` $\Lambda$. On the other hand, better semantic alignment is achievable with low `no_rep`, high $b$, and low $\Lambda$. Optimally tuned hyperparameters make Diverse Beam Search superior to standard Beam Search. In LDS, the $\mathcal{R}$ trend is unchanged, and $\Lambda$ has less impact on the syntactic surface. In particular, the effectiveness curve according to BERTScore changes radically, favoring high `no_rep`, low $b$, and low $\Lambda$. Moving

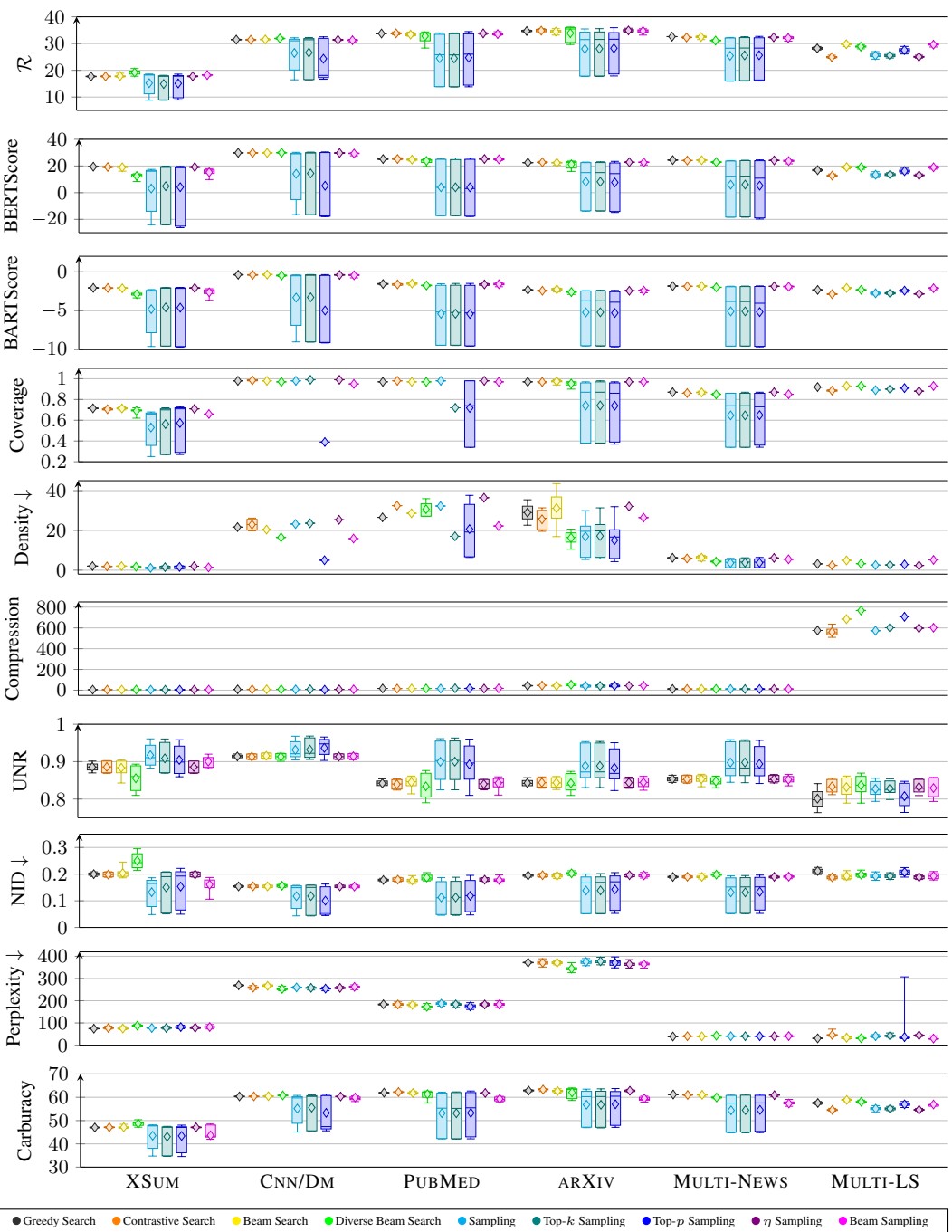

Figure 9: Graphical metric score distributions grouped by datasets and strategies.

to MDS, the general results of MULTI-NEWS improve with increasing $b$ and $\Lambda$. In all task families, there is a consistent upward trend in BARTScore as $b$ and no_rep values increase, except for XSUM.

**Sampling (Table 19)** In the case of SDS and MDS, the behavior is well-established, suggesting the adoption of high temperatures to maximize $\mathcal{R}$, BERTScore, and BARTScore. The only outlier is MULTI-LEXSUM, hypothesizing that it is due to the extreme length of its inputs, which causes inaccurate summaries to be generated. Perplexity varies across datasets and shows a weak correlation with changes in hyperparameters.

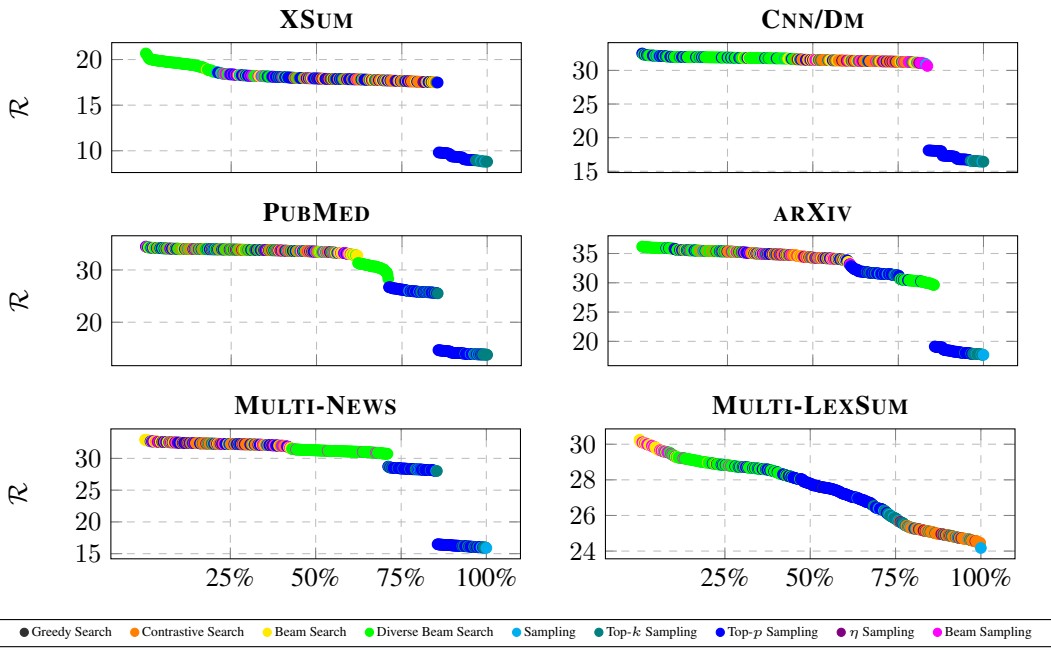

Figure 10: Per-dataset decoding strategy ranking according to $\mathcal{R}$.

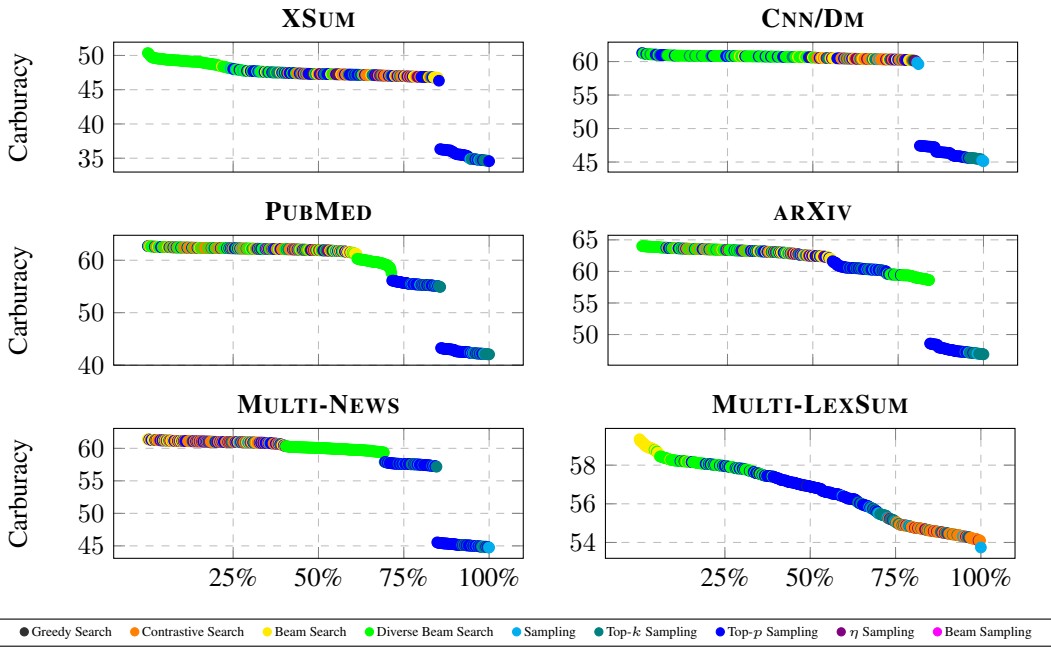

Figure 11: Per-dataset decoding strategy ranking according to Carburacy.

**Top-$k$ Sampling (Table 20) and Top-$p$ Sampling (Table 21)** Excluding MULTI-LEXSUM, $\mathcal{R}$, BERTScore and BARTScore improve as temperatures increase. When compared to temperature, `top-k`, `top-p`, and `no_rep` have very little effect on the metrics. The disparity between Top-$k$ and Top-$p$ Sampling mainly arises from Perplexity in longer inputs, favoring the latter approach.

**$\eta$ Sampling (Table 22)** The behavior of $\eta$ Sampling appears to be difficult to predict and control. As for BARTScore, results get better with high `no_rep`, while `cut_off` has a large impact only

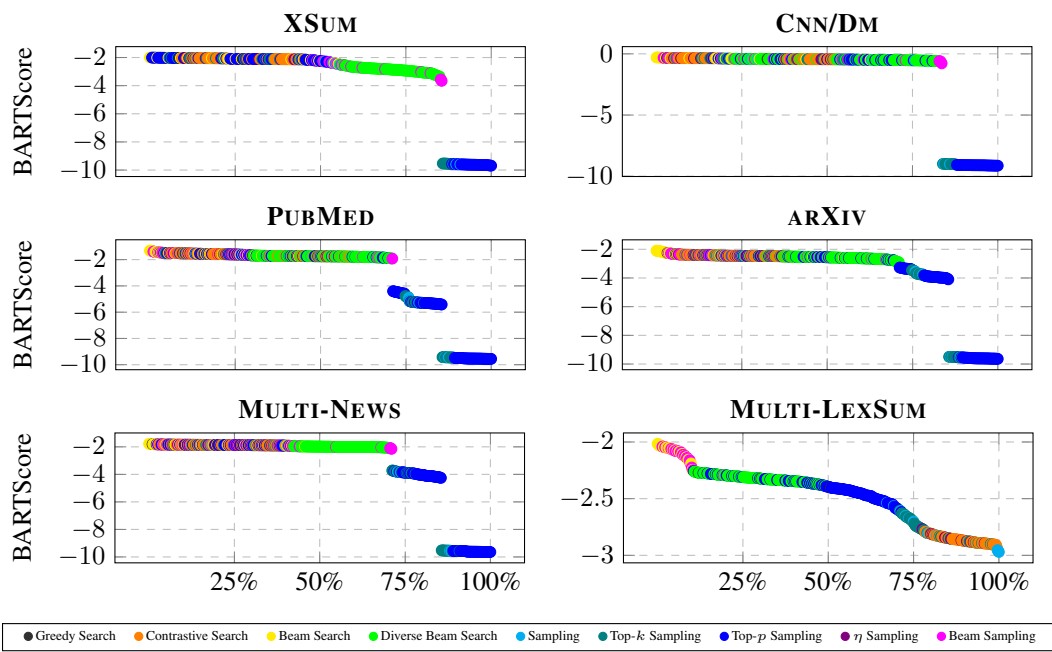

Figure 12: Per-dataset decoding strategy ranking according to BARTScore.

Table 12: Best $\mathcal{R}$ hyperparameters for decoding strategies across all datasets.

| Greedy Search | Contrastive Search | Beam Search |
|---|---|---|
| `no_repeat_ngram_size=3` | `no_repeat_ngram_size=5,`
`top_k=20,`
`penalty_alpha=0.2` | `no_repeat_ngram_size=3,`
`num_beams=9` |
| **Diverse Beam Search** | **Sampling** | **Top-$k$ Sampling** |
| `no_repeat_ngram_size=3,`
`num_beams=10,`
`diversity_penalty=0.2` | `no_repeat_ngram_size=5,`
`temperature=1.0` | `no_repeat_ngram_size=5,`
`temperature=1.0`
`top_k=50` |
| **Top-$p$ Sampling** | **$\eta$ Sampling** | **Beam Sampling** |
| `no_repeat_ngram_size=3,`
`temperature=1.0,`
`top_k=20,`
`top_p=0.4` | `no_repeat_ngram_size=5,`
`cut_off=0.002` | `no_repeat_ngram_size=3,`
`num_beams=10` |

on MULTI-LEXSUM. The rise in $\mathcal{R}$ often aligns with an increase in `no_rep`, while variations in `cut_off` exhibit a comparatively weaker impact.

**Beam Sampling (Table 23)** In the case of XSUM, there is a conspicuous positive correlation between $b$ and the metrics $\mathcal{R}$, BERTScore, and BARTScore (`no_rep` has a small influence). A similar pattern is observed for CNN/DM, where optimal performance is achieved with $b = 6$, followed by slight oscillations. In LDS, $\mathcal{R}$ and BERTScore have opposite trends in PUBMED and ARXIV. In MDS, $\mathcal{R}$ increases with $b$ and `no_rep`, with few exceptions in MULTI-LEXSUM.

## J  BEST HYPERPARAMETERS

Table 12 presents the hyperparameters that lead to the best $\mathcal{R}$ score (i.e., $\geq 0.9$ quantile).

## K  BEST DECODING STRATEGIES PER DATASET

Table 13 reports the configuration of the top-3 $\mathcal{R}$ decoding strategies per evaluated dataset.

Table 13: Hyperparameters for the top-3 $\mathcal{R}$ decoding strategies per dataset.

| Strategy | no_rep | num_beams | div_penalty | temperature | penalty_alpha | top_k | top_p | cutoff |
|---|---|---|---|---|---|---|---|---|
| **XSUM** | | | | | | | | |
| 1. Contrastive Search | 2 | - | - | - | 1.0 | 20 | - | - |
| 2. Top-$k$ Sampling | 2 | - | - | 0.9 | - | 50 | - | - |
| 3. Top-$p$ Sampling | 2 | - | - | 0.9 | - | 50 | 0.6 | - |
| **CNN/DM** | | | | | | | | |
| 1. Diverse Beam Search | 3 | 5 | 0.2 | - | - | - | - | - |
| 2. Top-$k$ Sampling | 3 | - | - | 0.9 | - | 50 | - | - |
| 3. Top-$p$ Sampling | 3 | - | - | 0.9 | - | - | 0.4 | - |
| **PUBMED** | | | | | | | | |
| 1. Contrastive Search | 5 | - | - | - | 0.8 | 30 | - | - |
| 2. Top-$p$ Sampling | 5 | - | - | 1.0 | - | 50 | 0.8 | - |
| 3. Beam Sampling | 5 | 3 | - | - | - | - | - | - |
| **ARXIV** | | | | | | | | |
| 1. Diverse Beam Search | 5 | 10 | 0.6 | - | - | - | - | - |
| 2. Top-$p$ Sampling | 5 | - | - | 1.0 | - | 20 | 0.6 | - |
| 3. $\eta$ Sampling | 5 | - | - | - | - | - | - | 0.0003 |
| **MULTI-NEWS** | | | | | | | | |
| 1. Contrastive Search | 5 | - | - | - | 0.2 | 30 | - | - |
| 2. Beam Search | 4 | 10 | - | - | - | - | - | - |
| 3. Beam Sampling | 5 | 9 | - | - | - | - | - | - |
| **MULTI-LEXSUM** | | | | | | | | |
| 1. Beam Search | 3 | 8 | - | - | - | - | - | - |
| 2. Diverse Beam Search | 3 | 7 | 0.2 | - | - | - | - | - |
| 3. Beam Sampling | 3 | 8 | - | - | - | - | - | - |

## L GUIDELINE

Based on average scores, ranking positions, and hyperparameter sensitivity, we suggest a small-size decoding space depending on the optimization target and AS type (Figure 13).

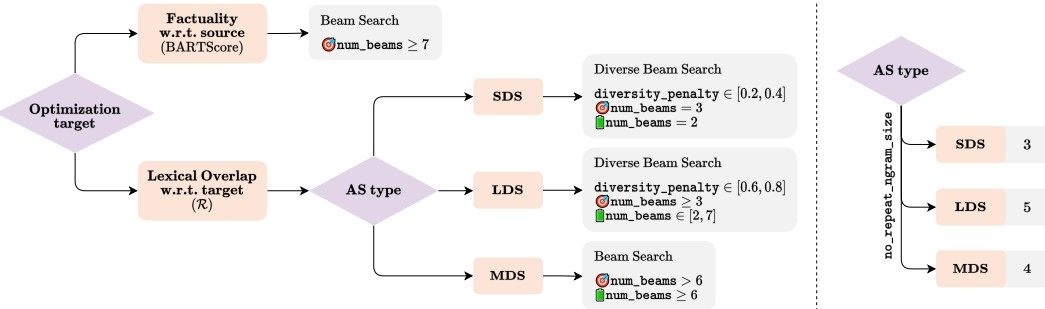

Figure 13: Visual guideline for decoding strategy selection. Optimization target = "What are you interested in?"; AS type = "What is your Abstractive Summarization scenario?". The hit symbol, when included, denotes hyperparameter value recommendations for achieving maximum effectiveness on the proxy metric. In contrast, the battery symbol signifies the optimal effectiveness-efficiency trade-off as proposed by Moro et al. (2023c).

## M GENERATION EXAMPLES

Table 14 shows some input-output examples in an XSUM instance.

## N LIMITATIONS AND FUTURE DIRECTIONS

**Additional Decoding Strategies** We focused on broadly applicable decoding methods that implement the same likelihood objective as the models. However, some recent strategies add substantial further assumptions, goals, non-metric-agnostic outputs, and task-specific constraints to generation. Among these, we mention FAME (Aralikatte et al., 2021), which dynamically biases the decoder to

Table 14: XSUM case study.

| Document |
|---|
| A penalty try and scores from alex goode and chris wyles gave sarries a 24-3 lead at half-time. The bonus point was wrapped up four minutes after the break as maro itoje crossed, shortly before richard wigglesworth touched down. Arthur aziza went over for oyonnax but schalk brits was awarded a late effort to complete the rout for saracens. Aziza is try was a deserved consolation for the french side, who were committed and spirited in their first ever home game in the champions cup, but were outclassed by a saracens team that sits nine points clear at the top of pool one. The premiership leaders dominated the scrum, until the introduction of a series of replacements in the second half upset their rhythm, and attacked with speed and purpose to signal their credentials as title candidates. New england head coach eddie jones will name his first squad next month, and there were plenty of performances from saracens' english contingent that would have caught the attention of the australian. But it was farrell who stood out with creativity and quickness that belied his reputation as a defensive, pragmatic fly-half. The england number 10 crashed through a gap in the oyonnax rearguard before swinging a looping pass out wide to wyles, who slipped in goode for the visitors' second try. and farrell was central to the next score, popping up an offload from the ground to allow will fraser to find the on-rushing wyles on his way to touching down under the posts. In fact, it may be 26-year-old fraser who gave jones the most to think about after this display. the uncapped open-side flanker was impressive with the ball in hand and could provide competition for the england number seven shirt, particularly with the future of england captain chris robshaw under scrutiny. Saracens can take a step closer to progressing when they host oyonnax in the reverse fixture next weekend. Saracens full-back alex goode said: we are really coming alive when we get the ball in good positions at the moment. We are using our strike runners very well. We are getting them in the game and we are picking them out. Guys like chris ashton are doing brilliantly in terms of the support lines they are running. We have had a young team over the last few years and we have grown together and added layers to our game. We have been close in this tournament before, we have our foundations, we are in a good place but we know there is a long journey ahead yet. Teams oyonnax: etienne; tian, bousses, taufa, codjo; clegg, cibray; tonga'uiha, maurouard, pungea; power, fabbri; ursache, missoup, wannenburg. replacements: lespinas for clegg (51), aziza for cibray (51), m. clerc for pungea (53), guillamon for fabbri (63), faure for wannenburg (60). Not used: bordes, rapant, gunther. saracens: goode; ashton, bosch, barritt, wyles; farrell, wigglesworth; m. vunipola, george, figallo; j. hamilton, itoje; rhodes, fraser, b. vunipola. replacements: ellery for wyles (55), hodgson for farrell (59), gill for m. vunipola (52), brits for george (52), du plessis for figallo (50), brown for itoje (63). not used: wray, de kock. Attendance: 11,400 referee: marius mitrea (italy). |

| Gold Summary |
|---|
| Saracens ran in six tries to thrash oyonnax and take control of their European Champions Cup group. |

| Greedy Search |
|---|
| Saracens eased to a bonus-point win over oyonnax in their Champions Cup opener at the Stade Gilbert Brutus in Toulon on Friday night to move to the top of Pool One in the tournament for the second year in a row, and extend their unbeaten run to seven matches.. the first time they have done so since 1999. and the third time since 2007. in that same match, when they were also beaten by the same French side in Paris. |

| Contrastive Search |
|---|
| Saracens continued their impressive start to the European Champions Cup with a bonus-point win over oyonnax at Stade Gilbert Brutus in Toulon on Friday night, to go top of Pool One.The home side ran in six tries against the French side, who were outclassed by a much-improved england side at the Madejski Stadium in Sydney on Saturday. the visitors have a game in hand in their next match against St Helens on Sunday. |

| Beam Search |
|---|
| Saracens cruised to a bonus-point win over oyonnax at the Stade Gilbert Brutus in Paris to move top of Pool One of the European Champions Cup, and set up a home quarter-final against Tonga or Fiji on 5 March, either side of the second leg of their tie at Twickenham on 7 March.. andrew farrell scored a hat-trick of tries as saracens thrashed the French side in their opening match of the tournament. |

| Diverse Beam Search |
|---|
| Saracens ran in five tries as they thrashed oyonnax to move top of their Champions Cup pool... the visitors were given a huge boost by the decision to award them a bonus-point win.. and. the first try was scored by alex goode..- the first of a number of tries that were scored by the visitors in the win over the French side.. alexgoode. and the second try was a penalty.. |

| Sampling |
|---|
| Will fraser scored a try and set up another as saracens ran in four tries to thrash oyonnax in their Champions Cup opener at the Stade Gilbert Brutus in Leuven. sacrificial lamb, that was how it was for the English side, who made light work of their first match in the competition since taking over from Australia's Eddie Jones in 2014.adiator farrell was the star of the show, running in two tries and setting up two others. |

| Top-$k$ Sampling |
|---|
| Saracens ran in six tries to thrash oyonnax in their opening match of the European Champions Cup at the Stade Gilbert Brutus in Paris on Friday night, and put themselves in a strong position to progress to the quarter-finals of a tournament they have won three times in the last four years, with the highest ever total of 14 tries scored by a British and Irish Lions team in such a fixture. the game was played in English, not French, but the score was never in doubt. |

| Top-$p$ Sampling |
|---|
| Saracens eased to a bonus-point win over oyonnax in their Champions Cup opener at the Stade Gilbert Brutus in Toulon on Friday night, to move to the top of Pool One with one game to play in the competition, and to give themselves a chance of progressing to the quarter-finals of the European Champions Cup in New Zealand on Sunday, 3 January, with a win over Tonga in their second game of the same fixture on the same day. |

| $\eta$ Sampling |
|---|
| Saracens eased to a bonus-point win over oyonnax in their Champions Cup opener at the Stade Gilbert Brutus in Toulon on Friday night, to move to the top of Pool One. the premiership leaders dominated the scrum, until the introduction of a series of replacements in the second half, and have the chance to reach the quarter-finals of the competition for the first time since 2007. hasan farrell scored a hat-trick of tries. |

| Beam Sampling |
|---|
| Saracens eased to a bonus-point win over French champions oyonnax at Murrayfield in their opening Champions Cup match of the 2016-17 season.arian farrell scored a hat-trick of tries for the visitors, and they extended their lead at the top of Pool One. eddie jones' new side got their first win under way with victory over the French side at Twickenham. FAA. Wakefield. Omnia.aii. Mahlon. Raganski. |

proactively generate summary tokens that are topically similar to the input, and PINOCCHIO (King et al., 2022), which restricts Beam Search only to consider tokens that are likely to be supported by the source text. Future work should investigate their inclusion in the analysis.

**Additional Models**    We conducted a diligent model selection process to identify representative state-of-the-art encoder-decoder solutions within each AS family (short, long, multi-document). Chosen models have gained widespread recognition and feature fine-tuned checkpoints from reputable sources (Meta and AllenAI). In the scope of this study, our emphasis was on evaluating the models most commonly used by the AS community, rather than those that might gain prominence in the future. These models belong to a million parameter scale or hardly overcome 10B, a lower boundary often cited when discussing LLMs (Zhao et al., 2023). Extending our study to LLMs such as GPT-3 (Brown et al., 2020) and open source alternatives would be noteworthy. However, we underline that resorting to multiple LLMs would be essential to avoid reaching decoding-level conclusions solely based on the behavior of a particular model (and not an entire spectrum of modeling techniques), an aspect frequently overlooked in previous publications (Meister et al., 2022b; Su et al., 2022; Su & Collier, 2023). Furthermore, limiting the LLM analysis to a subset of AS types would not be a sound assessment choice. Also, note that, according to our experimental tests, moving from the 440M parameters to the 7B versions of BLOOM (Scao et al., 2022), FALCON (Almazrouei et al., 2023), or LLAMA-2 (Touvron et al., 2023) averagely increases the decoding time per instance by a factor of x150. As a result, this would inadvertently multiply the required computational needs and runtimes. We thereby believe that incorporating LLMs and decoder-only architectures deserves an ad hoc extension. We know that the choice of the model may introduce bias in interpreting the results between different families. However, each AS type notoriously requires specific adaptions. Thus, we opt for the best representative models for each type, following BART-inspired architectures.

**Additional Datasets**    Regarding testbeds, we carefully considered two well-known representative public datasets per AS type; further analysis should contemplate additional datasets from each AS family (e.g., BILLSUM (Kornilova & Eidelman, 2019a) for SDS and GOVREPORT (Huang et al., 2021b) for LDS). Future work should investigate other AS fields, such as dialogue summarization (Feng et al., 2022), and other NLG tasks, such as question answering.

**Multi-Lingual Resources**    The experimental analysis is focused solely on English. However, it is important to recognize that this choice was influenced by several factors, including the availability of datasets and fine-tuned models in the AS settings analyzed. English serves as a lingua franca in many domains, achieving a broader impact and enabling a substantial body of research to be conducted effectively due to the wealth of resources. Shifting towards multi-lingual insights necessitates an entirely new set of models and datasets for each AS type, thus doubling the number of runs and computational days. We plan to extend our analysis to multi-lingual settings in future work.

**Increased Sample Size**    We recognize that we used a limited set of samples in comparative experiments. Testing all delineated configurations on all entire datasets would inadvertently multiply the required computational needs and runtimes, becoming unsustainable with our hardware configuration (Appendix E). To mitigate this problem, we performed stratified sampling, reflecting the population rightly and avoiding common pure random sampling.

**Novel Decoding Strategy**    This research centers on a thorough comparison of existing decoding methods alongside exhaustive hyperparameter permutations and evaluation axes. The goal is to grasp the impact of well-established decoding strategies on AS tasks from a neutral point of view and provide fresh insights to the community. In contrast to previous work, such as Top-$p$ Sampling (Holtzman et al., 2020) and Contrastive Search (Su et al., 2022), we avoid presenting another decoding technique with a shallow evaluation related to affected quality attributes (e.g., naturalness and diversity for Top-$p$, coherence for Contrastive Search). For these reasons, the introduction of a new heuristic falls beyond the scope of our study in favor of a horizontal assessment on an unprecedented scale.

**Diversity Metrics**    We acknowledge that there are multiple ways to summarize a document and that the lack of diversity in decoded solutions is crippling in the NLG sector, which highlights a relevant development for our paper. However, the quality and diversity criteria are not equally important in all tasks, where improving one often comes at the cost of the other (Aralikatte et al., 2021). According

to Meister et al. (2022b), we argue that AS goals place a higher premium on accurate outputs than alternative ones. This factor is opposite to open-ended domains and tasks such as dialogue generation, aiming to maintain an enjoyable conversation with a human partner, avoiding repetitions.

**Unified Metric** Unlike other benchmarks such as GLUE (Wang et al., 2018) and SuperGLUE (Wang et al., 2019), this study uses metrics to evaluate orthogonal quality dimensions in generated summaries, from prediction-target semantic similarity to source-target compression and actuality. These aspects encompass different goals, boundaries, and interpretation keys, which make it not viable or sound to establish a single overall score. It is worth noting that, based on our analysis of the literature, no previous research has computed a set of 10 distinct NLG metrics to assess artificial summaries, as we have in PRISM. We hope that our work will help researchers grapple with the identification of the finest decoding options for AS and set the stage for a new wave of heuristics.

Table 15: Hyperparameters effect grouped by datasets – Greedy Search.

Table 16: Hyperparameters effect grouped by datasets – Contrastive Search.

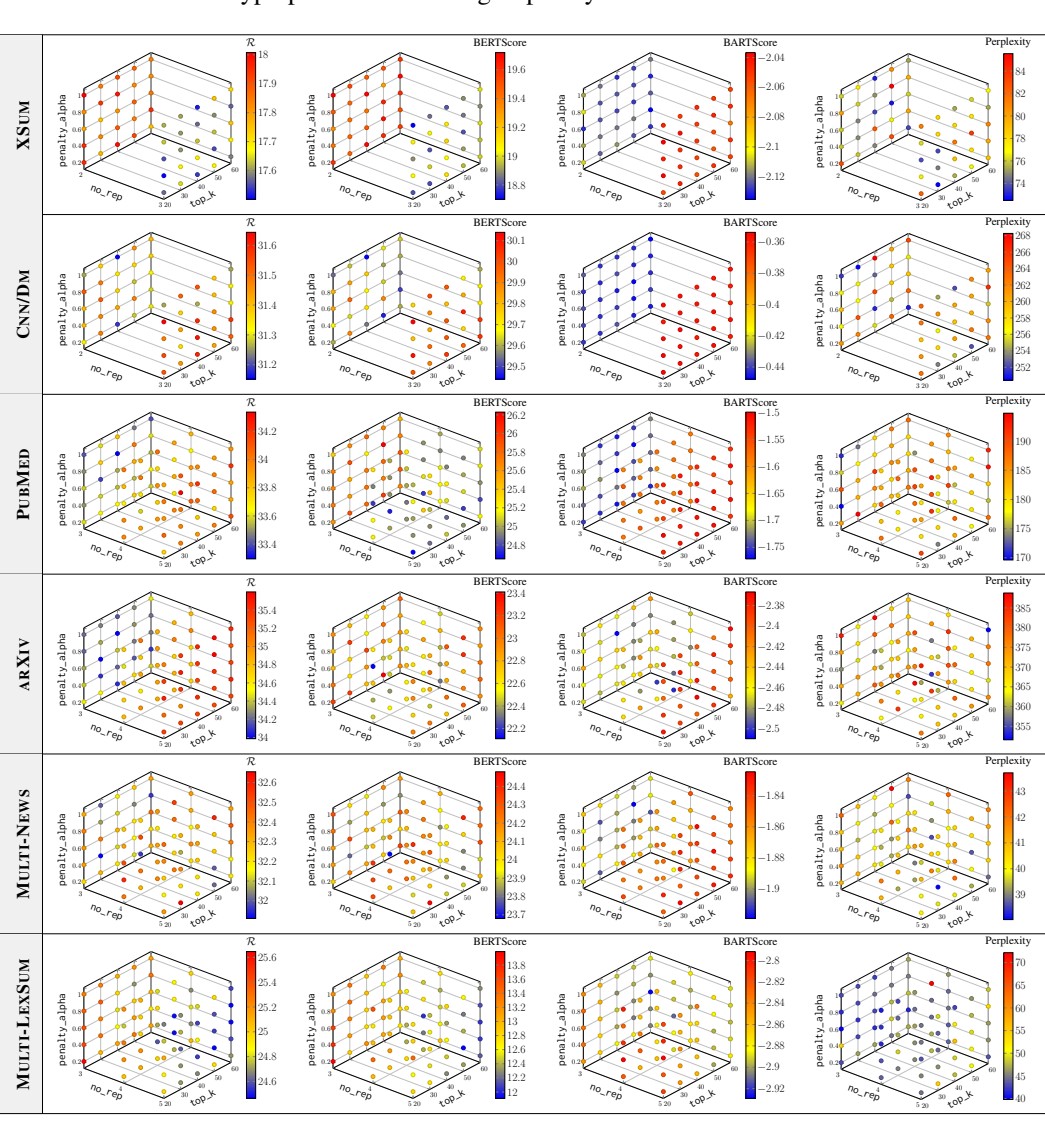

Table 17: Hyperparameters effect grouped by datasets – Beam Search.

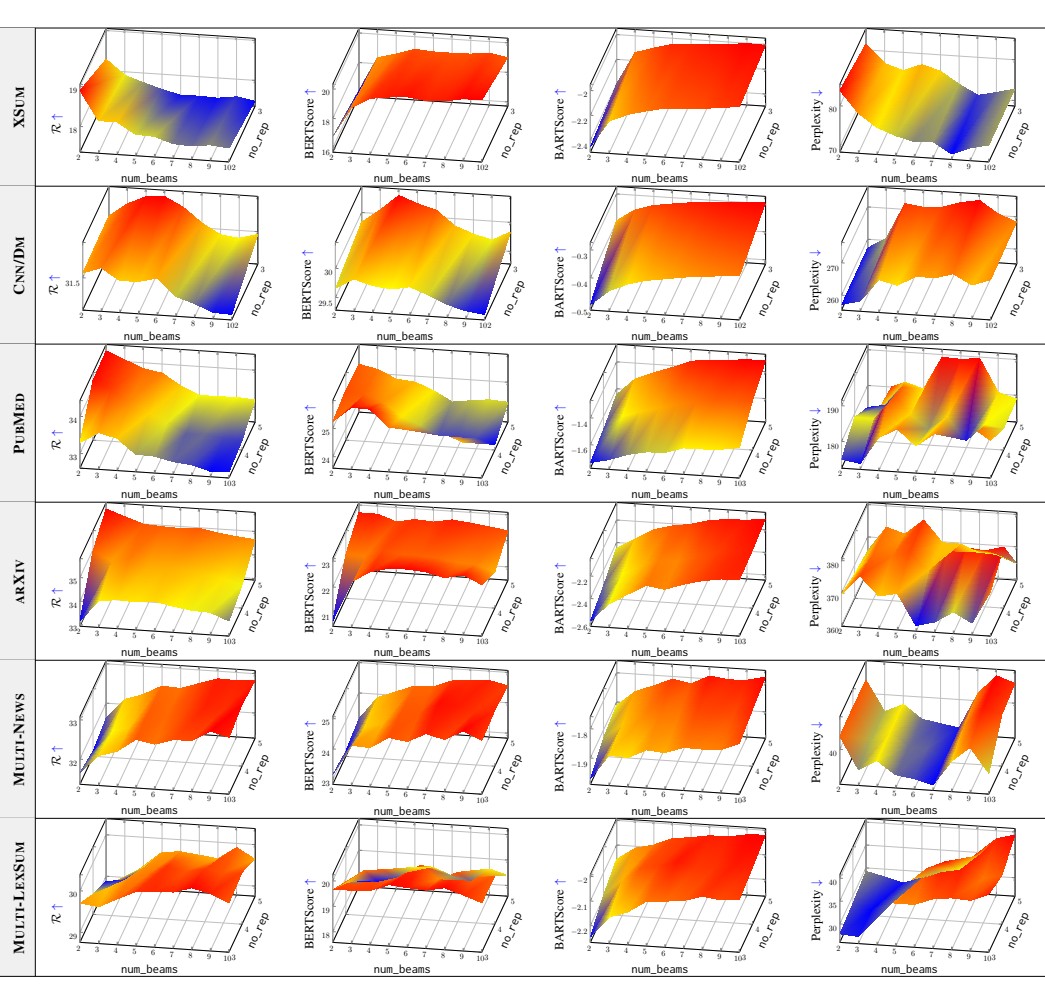

Table 18: Hyperparameters effect grouped by datasets – Diverse Beam Search.

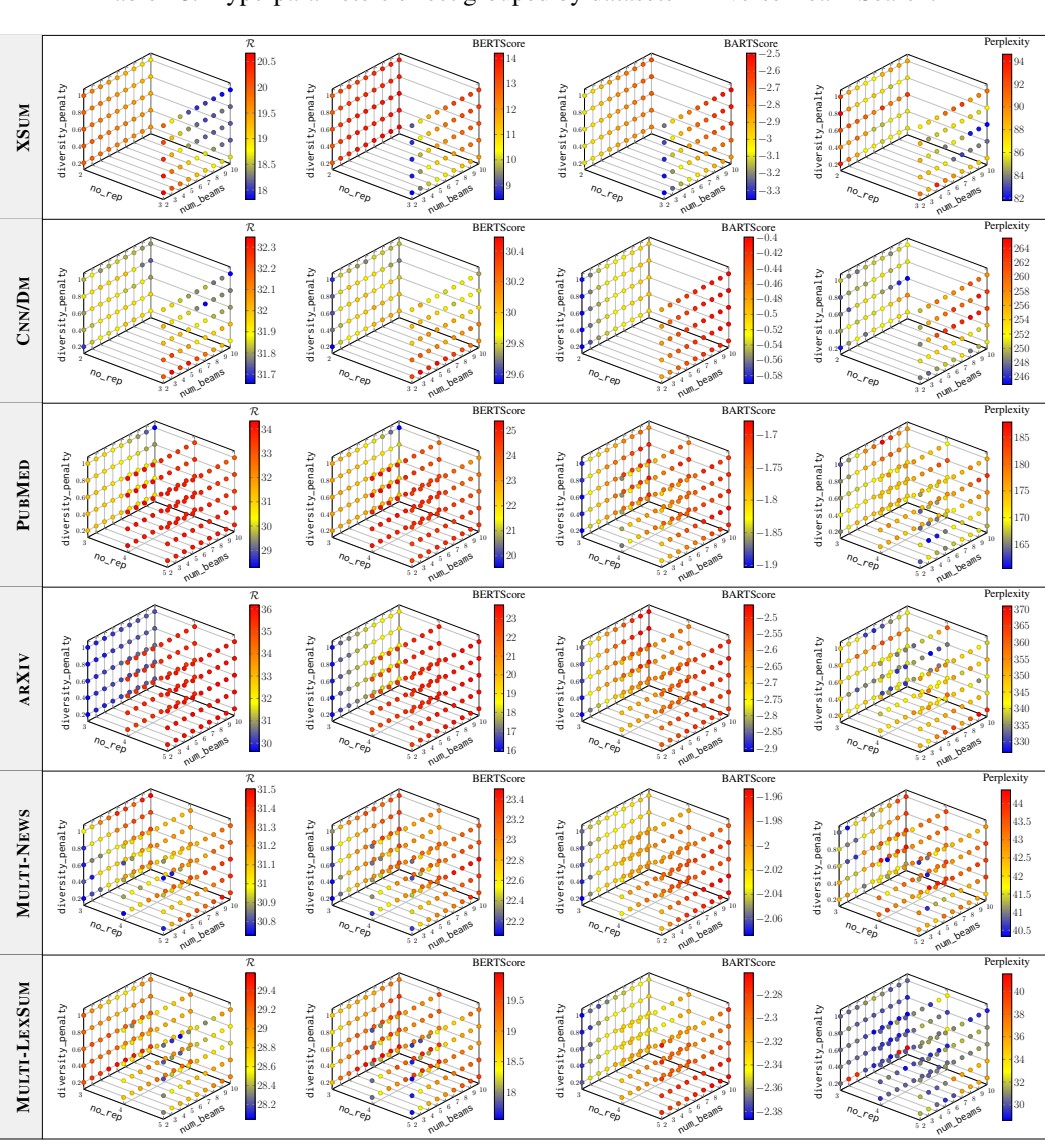

Table 19: Hyperparameters effect grouped by datasets – Sampling.

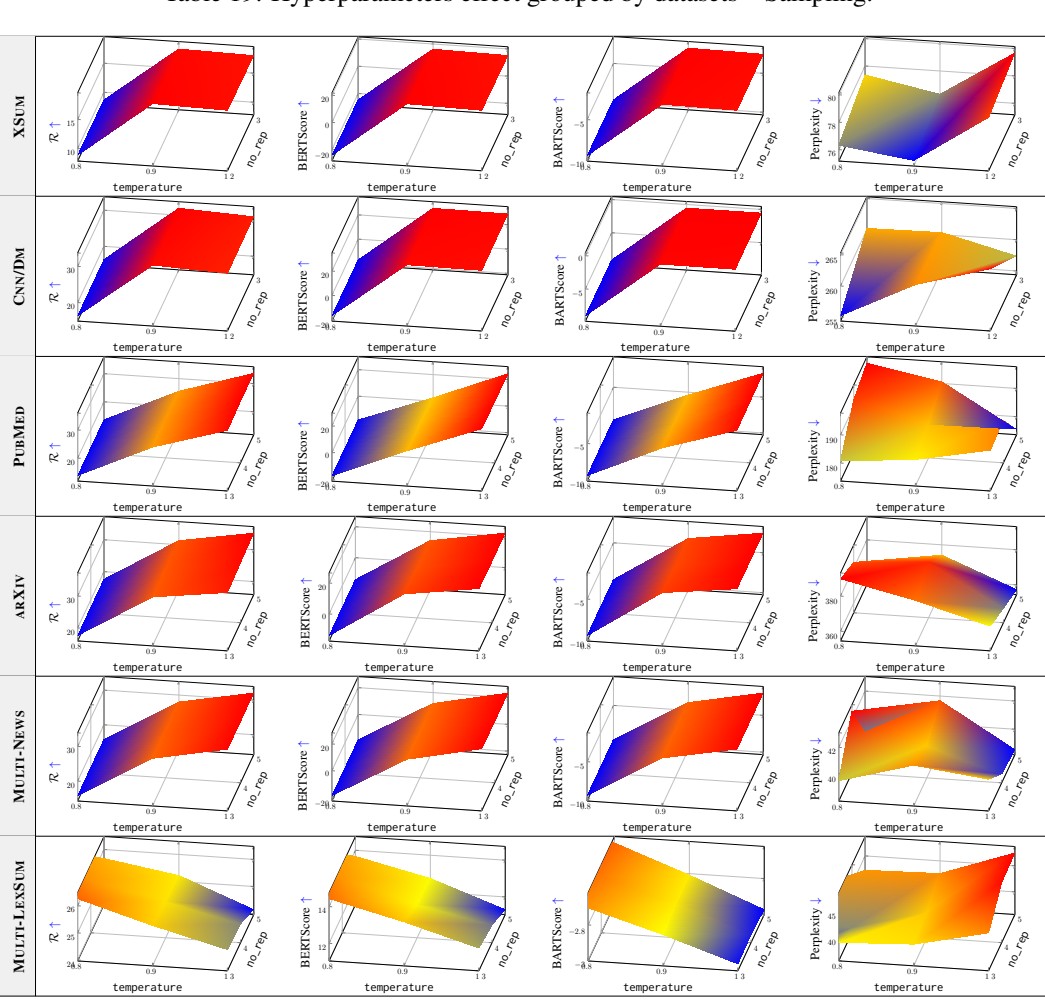

Table 20: Hyperparameters effect grouped by datasets – Top-$k$ Sampling.

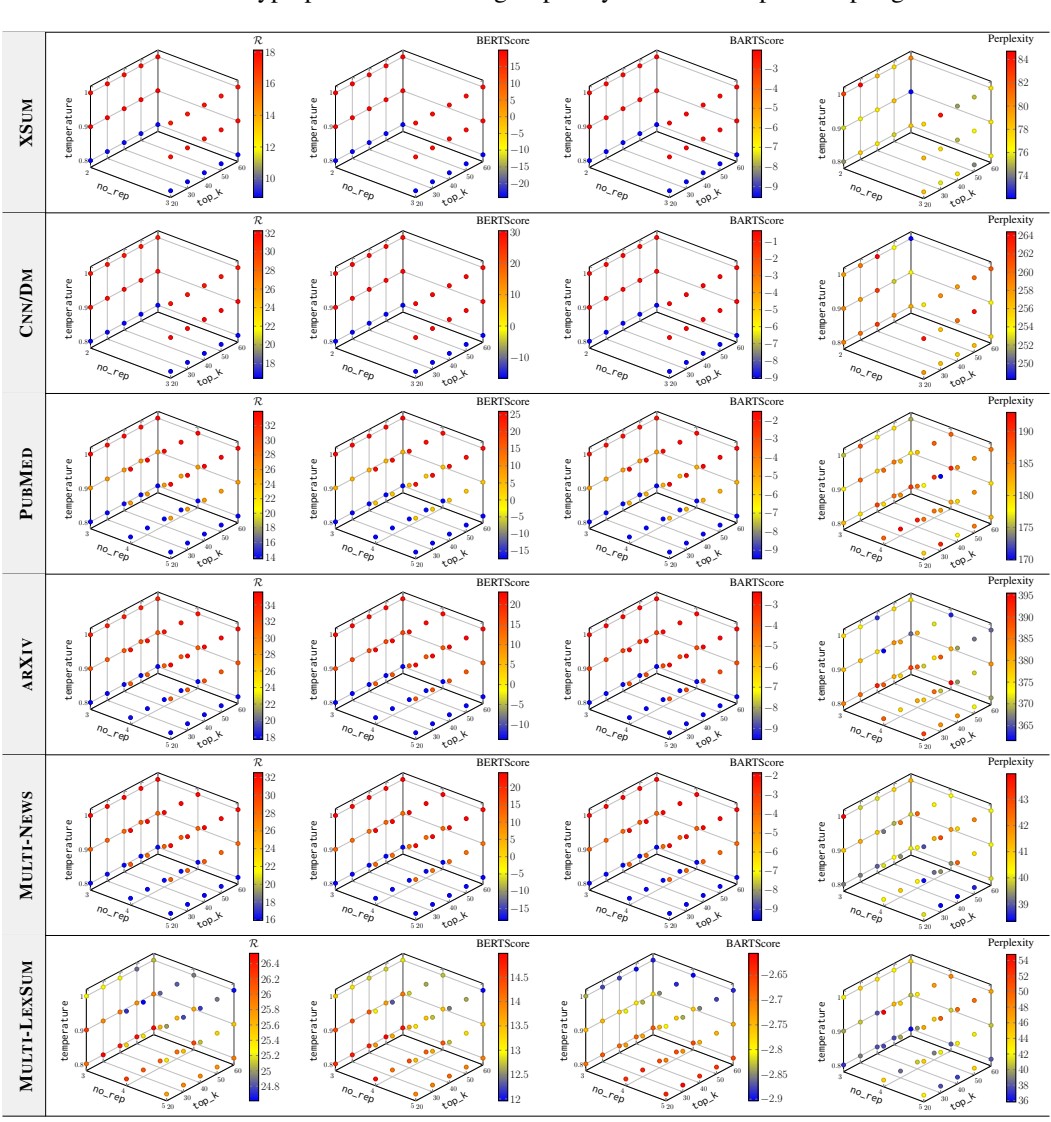

Table 21: Hyperparameters effect grouped by datasets – Top-$p$ Sampling.

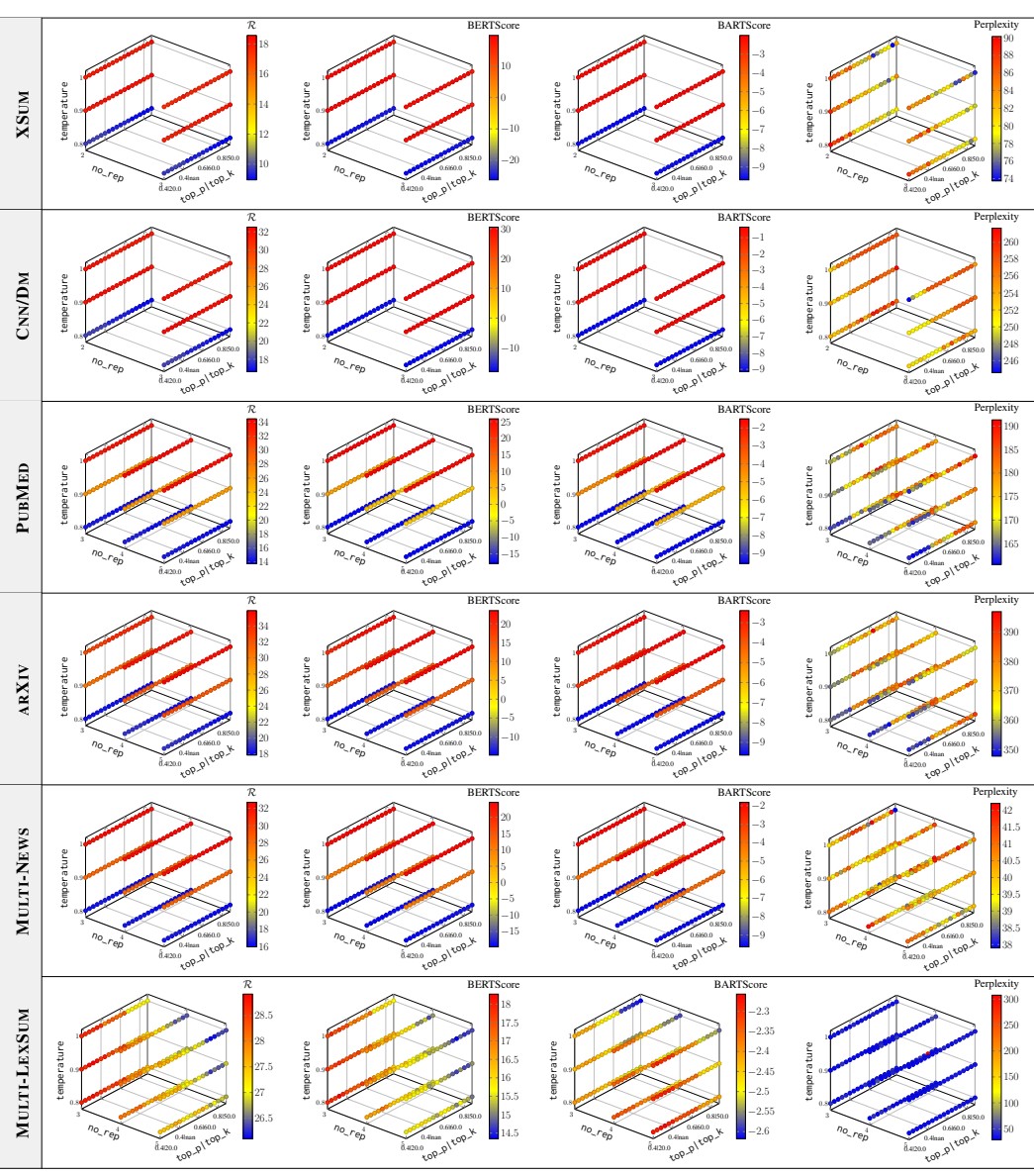

Table 22: Hyperparameters effect grouped by datasets – $\eta$ Sampling.

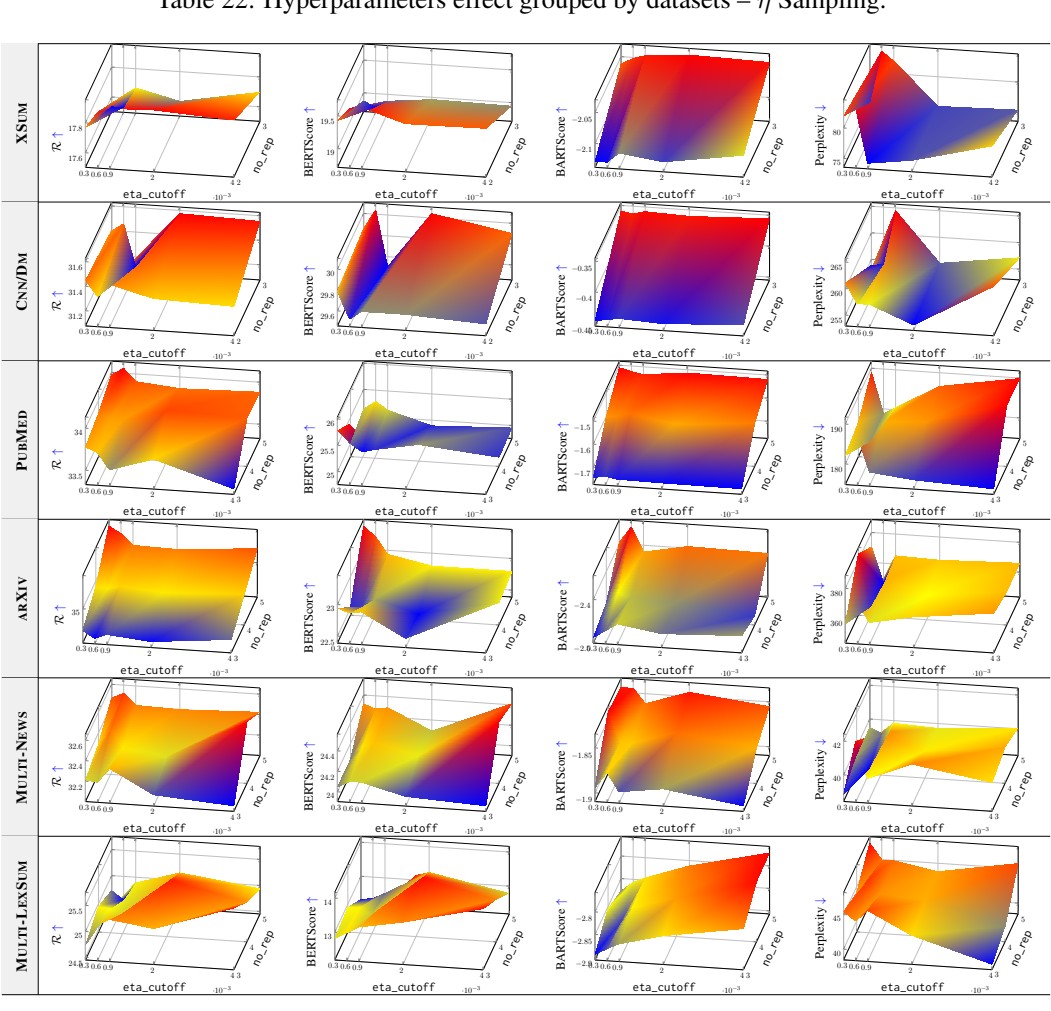

Table 23: Hyperparameters effect grouped by datasets – Beam Sampling.

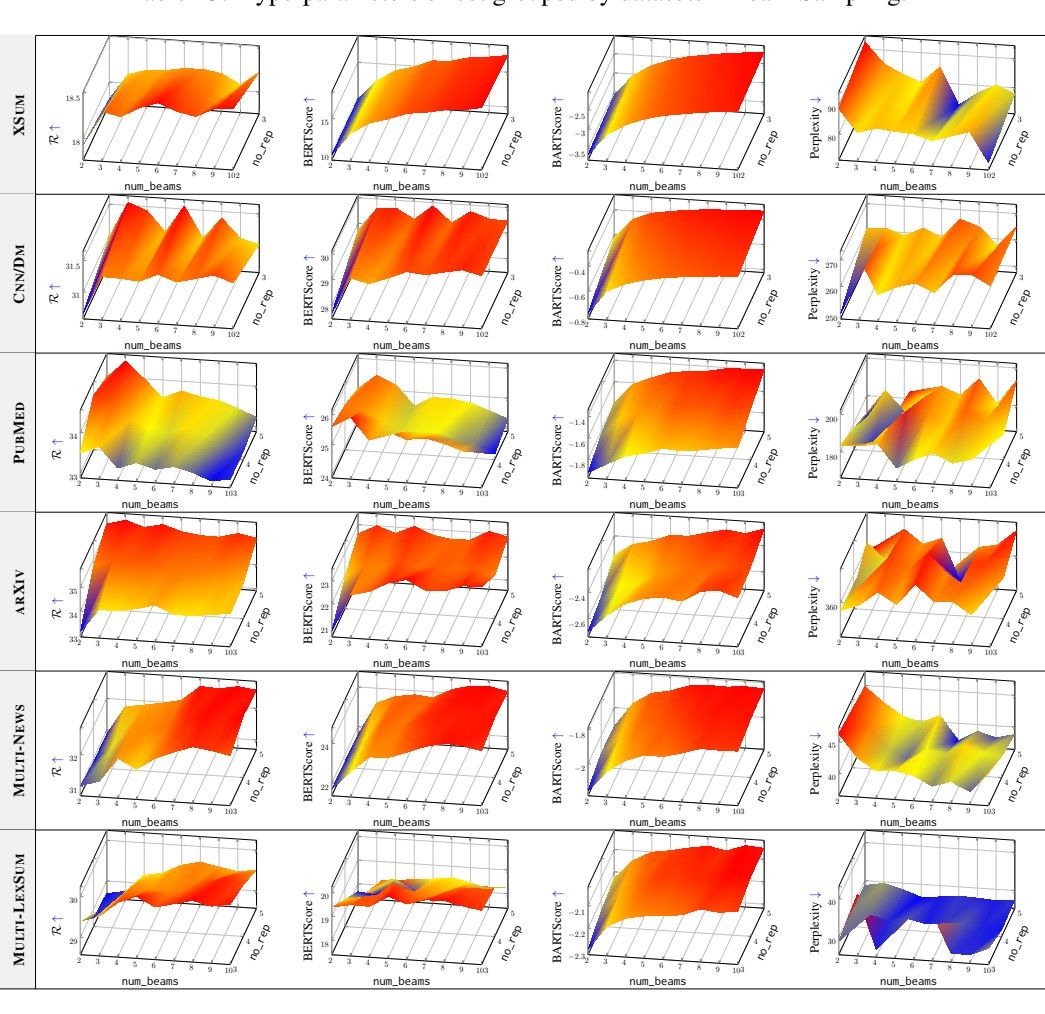

