# OpenReview forum: "Abstractive Summarization through the PRISM of Decoding Strategies"
_ICLR.cc/2024/Conference — Submitted to ICLR 2024_

### Official Review · Reviewer_BSXa · 2023-10-13

**Soundness:** 3 good
**Presentation:** 3 good
**Contribution:** 2 fair
**Rating:** 5
**Confidence:** 4

**Summary:**

This paper explores over 2,500 combinations of three widely-used million-scale autoregressive encoder-decoder models, across six datasets and nine decoding settings. The research reveals that optimized decoding choices can significantly boost performance.

**Strengths:**

1. The authors conduct extensive experiments across various combinations and datasets.

2. The paper provides visualized and in-depth analysis.

**Weaknesses:**

1. Given that there's no universal strategy for abstractive summarization, the practical value of the experiments is questionable. For the investigated question like "Which hyperparameter values best suit a particular AS quality attribute?", the answer is "Not all quality attributes are easy to temper at decoding time.", which seems ambiguous and might not guide future work.

2. Extending the method to LLM models would have made the findings more impactful in the era of LLM.

**Questions:**

none

---

> ### Author Response · Authors · 2023-11-23
> **Thank you for your review!**
>
> We appreciate your constructive feedback.
>
> **(1) Questionable value of the experiments**
> > (review quote) Given that there's no universal strategy for abstractive summarization, the practical value of the experiments is questionable. For the investigated question like "Which hyperparameter values best suit a particular AS quality attribute?", the answer is "Not all quality attributes are easy to temper at decoding time.", which seems ambiguous and might not guide future work.
> - In response to your feedback, we revised and expanded our analysis, introducing per-dataset rankings (Appendix H) and new findings (Section 5). We identified prevalent performance patterns and established a set of recommendations to better assist practitioners in real-world summarization use cases. We attached a visual and easy-to-follow guideline to the conclusion (Appendix L), driving the selection of valuable decoding spaces (i.e., strategy + hyperparams) depending on the optimization target and AS type. We completely reworked the discussion of our results (Section 5), providing a more precise and in-depth dissection of quantitative and qualitative findings thanks to content re-organization, better answering each research question. We enhanced the presentation quality of the entire paper, including the Appendix, prioritizing rigor and clarity in our statements. The absence of a universal decoding strategy does not affect the validity of our analysis, the practical value of the experiments, or the potential to guide future work. To our knowledge, we are the first to (i) thoroughly examine the decoding results for AS in its various forms with respect to the input length and n-arity, (ii) present findings based on literature-supported grid search tuning for decoding hyperparameter spaces, and (iii) release a pioneering decoding-oriented dataset for AS to open new research directions. The findings unveiled by our research are several and include: (a) the shortfall of a one-size-fits-all strategy for AS (as underlined by the reviewer), (b) the huge effect of decoding strategies on final AS metric scores (as emphasized by the reviewer in the strength points), (c) the effectiveness—efficiency trade-off of each decoding strategy, (d) the high sensitivity of stochastic methods to hyperparameter calibration, (e) the identification of quality attributes controllable at decoding time, (f) the ubiquitous nature of redundancy in AS, (g) the unexpected effectiveness of well-tuned stochastic methods for concise target summaries, (h) the preference of deterministic heuristics when the target is long, (i) the hard balance between fluency and recall, precision, faithfulness, (j) the highly competitive level of summaries produced by diverse strategies after optimal tuning.
>
> **(2) Lack of Large Language Models**
> > (review quote) Extending the method to LLM models would have made the findings more impactful in the era of LLM.
> - Please refer to our response to _[Reviewer Eedp]_, point (2).
>
> **(3) Limited contribution**
> > (review quote) Contribution: 2 (fair)
> - We appreciate the review and the effort put into the evaluation. However, we respectfully disagree with the assigned score, as we believe that it does not accurately reflect the depth of our contributions. We are not convinced that reported concerns adequately support a score of 2, the lowest received. We would be grateful for further clarification on specific points to address any perceived shortcomings more effectively.
>
> Further elaboration on the introduced changes can be found in the general comment.

---

### Official Review · Reviewer_rLZb · 2023-10-29

**Soundness:** 2 fair
**Presentation:** 3 good
**Contribution:** 3 good
**Rating:** 6
**Confidence:** 3

**Summary:**

The paper conducts a large-scale study on the impact of decoding strategies on abstractive summarisation.
The authors investigate 9 decoding strategies (greedy search, contrastive search, beam search, diverse beam search, sampling, top-k sampling, top-p nucleus sampling, beam sampling, eta sampling) on 6 datasets (XSum, CNN/DM, PubMed, arXiv, Multi-News, Multi-LexSum), combined with 3 seq2seq models. In addition to human evaluation on 4 dimensions (recall, precision, faithfulness, fluence), the authors employ 10 automatic metrics (rouge, BERTScore, perplexity, coverage, density, compression, unigram n-gram ratio, normalized inverse of diversity, BARTScore, and Carburacy) and a few observations are provided.

**Strengths:**

* A large-scale study on multiple decoding strategies
* A resource that enables further studies on abstractive summarisation regarding evaluation metrics.
* The paper is easy to follow, except some figures are hard to interpret

**Weaknesses:**

It is hard to gain much insight from the results: sometimes it is unclear whether the authors are commenting on decoding strategies, datasets, or evaluation metrics.

**Questions:**

A: can you elaborate on the human evaluation dimension of faithfulness, especially how you instruct annotators to differentiate it from precision and recall?
B: it will be appreciated if some practical guidelines can be summarized

---

> ### Author Response · Authors · 2023-11-23
> **Thank you for your review!**
>
> We appreciate your comments.
>
> **(1) Need for more insightful results**
> > (review quote) some figures are hard to interpret
> > (review quote) It is hard to gain much insight from the results: sometimes it is unclear whether the authors are commenting on decoding strategies, datasets, or evaluation metrics.
> > (review quote) it will be appreciated if some practical guidelines can be summarized
> - We acknowledge your concerns and are pleased to receive feedback on results presentation. We have invested considerable time and effort on this aspect because of the high complexity originating from the numerous analysis dimensions. Recognizing the importance of clarity and interpretability in such a comprehensive study, we have taken significant steps to address this issue. In response to your observations, we implemented the following changes: (i) we refined the graphical representation of averaged metric results in the radar plots (Figure 3), also introducing its tabulated version (Table 11); (ii) we provided a more detailed discussion of quantitative and qualitative results, enriching the answer to every research question (Section 5); (iii) we included per-dataset ranking analyzes, making inter-dataset patterns easily perceptible (Appendix H); (iv) we assembled an easy-to-follow guideline summarizing key recommendations tailored to user needs (Appendix L); (v) we elevated the overall presentation quality of the entire paper. We trust that these adjustments will address the concerns raised and provide a more insightful and accessible presentation of our findings.
>
> **(2) Annotation instructions for human evaluation**
> > (review quote) can you elaborate on the human evaluation dimension of faithfulness, especially how you instruct annotators to differentiate it from precision and recall?
> - We have added more details about the instructions furnished to annotators in Appendix F. In particular, we have incorporated a highly comprehensible example (provided within the instructions) aimed at elucidating the distinctions among the evaluation dimensions of Recall, Precision, and Faithfulness. As highlighted in Section 4.4, we note that distinguishing Faithfulness from Precision and Recall is straightforward. This is because Faithfulness involves comparing the prediction with the input document, as opposed to the gold target summary as done for Precision and Recall.
>
> **(3) Limited soundness**
> > (review quote) Soundness: 2 (fair)
> - We appreciate the effort to make the review. However, we respectfully disagree with the assigned score, as we believe that it does not accurately reflect the technical soundness of our methodological approach. We are not convinced that the concerns and weaknesses reported adequately support a score of 2. We would be grateful for further clarification on specific points to address any perceived shortcomings more effectively (if not already resolved by the novel updates in the paper).
>
> Further elaboration on the introduced changes can be found in the general comment.

---

### Official Review · Reviewer_8Gmg · 2023-10-30

**Soundness:** 2 fair
**Presentation:** 2 fair
**Contribution:** 3 good
**Rating:** 5
**Confidence:** 4

**Summary:**

This paper presents an extensive study of the effect of various decoding methods on the quality of generated summaries on the task of abstractive written text summarization. The results cover three encoder-decoder transformer models with finetuned weights matched to six public abstractive summarization datasets under three categories: short text, long text, and multi-document. Explicit hyperparameter tuning was done on each (decoding strategy, dataset) pair, and the variation was evaluated by a multitude of automatic text evaluation metrics (lexical and semantic) as well as human evaluation along four quality dimensions (precision, recall, factuality, and fluency). The general findings confirm a lack of dominating decoding strategy for all tasks studied, and show that hyperparameter settings are also task/decoding strategy specific with potentially significant variation in calculated metrics between optimal and suboptimal hyperparameters.

**Strengths:**

Originality:
One of the biggest strength of the paper is in its scope of decoding methods (nine) and datasets (six). Although the study of effect of decoding methods on the quality of generated summaries by summarization models has been gaining momentum in the research community, a study of the scale presented in this paper has yet to be published.

Quality:
The results presented in the paper cover a wide range of research parameters (task, dataset, hyperparameter, strategy, evaluation metrics, automatic vs human evaluation). The authors have also been quite thorough in including detailed explanation on the data selection, preprocessing, and evaluation. The power analysis in Appendix D is a welcoming addition to the paper, lending reference to robust statistical analysis on the potential reliability of the findings in the paper.

Clarity:
The paper is generally well-organized, with thorough explanation on the experimental settings for clear reproducibility test.

Significance:
Another major strength of the paper is the significance of the direction of the research. Evaluation metrics and decoding strategies are two main aspects in abstractive summarization that are relatively underexplored compared to research on newer summarization models or new datasets. However the decoding methods undoubtedly play an influential role in the task of summarization and text generation in general. There are arguments on the misalignment between cross-entropy based training objective and actual quality measures of interest, and the success of many reranking methods also reflects the suboptimal strategy of blindly following a single decoding strategy. Therefore, the study done in this paper can provide referential values for any related and future research in summarization that aims at improving quality of the generation.

**Weaknesses:**

Major weakness of the paper is the presentation style of main results in Figure 3, and some inconsistency between Figure 3, the writing, and Figure 8 that may bring some of the quantitative findings into question:

1. Because of the density of information and the lack of differentiability in the chosen colors for different decoding strategies, Figure 3 is very hard to follow. For example, Greedy Search and Beam Sampling are both presented with "reddish" color, and the reviewer finds it very difficult to identify which is which in the plot. Additional comments on the figure are presented as questions in the "Questions" section

2. Because of the lack of interpretability of Figure 3 and lack of tabulated data for the results, it is hard for the reviewer to confirm the quantitative differences referred to in the Results section (e.g.,  +48% UNR avg. score mentioned between short and long summaries).

3. Contrary to the discussion on the relative advantage of certain decoding strategies over other in section 5.1, the impression of the reviewer from Figure 8 is that
(i) sampling, top-k, and top-p sampling clearly have larger sample variance than all other strategies, even after hyperparameter tuning, on five out of six datasets;
(ii) the behavior of all decoding methods changes significantly on Multi-LS dataset, showing larger scale of variation in average score and much smaller sample-specific variance;
(iii) apart from the three sampling strategies mentioned in (i), the remaining methods on all datasets except Multi-LS seem to be consistently on par with each other as evaluated by most of the evaluation metrics

these observations may contradict the quantitative findings in section 5.1, and there seems to lack specific discussion on those arguably more obvious patterns than the ones presented in Figure 3 and discussed in the text.

Another weakness is the generalizability of the findings. The reviewer agrees with the authors that the lack of a universal strategy for optimal decoding is substantively supported by the results, but apart from that the paper seems to be lacking in identifying other general trend or patterns in combination of the research parameters (decoding strategy, task, model, hyperparameters). This limits the value of the results to be observational rather than inferential.

Other problems regarding writing:
1. **Elevating Fluency undermines Recall, Precision, and Factuality.** in Section 5.2 is not a rigorous statement based on the results, what the authors have observed is a negative correlation between fluency and other quality factors as evaluated by human, but the phrasing of the finding turns such a correlation into a misleading causal statement.
2. It is not rigorous to draw conclusion on "factuality", "factual flaws", or "hallucination" solely based on BARTScore (e.g., Section 5.1, **stochastic vs deterministic**), which is at best a proxy for those quality measures. Please at the very least state in the main text that the studied automatic metrics are "proxies" of the quality dimensions of interest.
3. Typo: Table 4, shouldn't UNR be "Unique N-gram Ratio", not "unigram n-gram ratio"?

**Questions:**

1. In Figure 3, why are some of the metrics score concentrated towards low value (center of each circle, e.g., Perplexity on Multi-News)? The  reviewer may have misunderstood the presentation here, but if all scores are rescaled to [0, 1] based on min and max of each score, shouldn't there be a point on the outer perimeter for every metrics on every dataset (that is, the decoding method with the maximum score in a metric on a dataset)?
2. Any special reason why y-axis in Figure 4 is not presented on a linear scale?
3. Can you elaborate, maybe in the appendix, more details on the training regimen used in the human evaluation? Were the annotators given any examples or detailed explanation on each quality dimension? How did you guarantee that the source document would be viewed by the annotator? Or were the annotators already familiar with the research objectives so that training was minimal?
4. How is Kendall's tau calculated for human evaluation results? Was it calculated between two annotators across all pair selection results, or was a ranking of all decoding strategies first determined based on all selections from one annotator, and then tau calculated between two rankings? Why are the reported values so low and even negative on some datasets? If the annotators disagree significantly in their preference, wouldn't that seriously limit the possibility of drawing any robust conclusion based on their evaluation results?
5. Why not consider more semantic similarity metrics (e.g., NLI-based evaluation, see https://aclanthology.org/P19-1213.pdf for example) or NER based metrics? Apart from BertScore and BartScore, almost all other metrics used are lexical based (n-gram or character overlap). It's hard to say if the variation observed in those metrics truly reflect variation in qualities of the generated summaries (and based on Figure 8, it's hard to say if significant variation has been observed).

---

> ### Author Response · Authors · 2023-11-23
> **Thank you for your review!**
>
> We sincerely appreciate your constructive feedback.
>
> **(1) Need for presentation improvements**
> > (review quote) Major weakness of the paper is the presentation style of main results in Figure 3, and some inconsistency between Figure 3, the writing, and Figure 8 that may bring some of the quantitative findings into question.
> - Our decision to utilize radar plots in Figure 3 was driven by the belief that they offer a more insightful representation than tabular formats, particularly in addressing our research questions. The colored area surfaces in the radar plots facilitate quick and comprehensive visual comparisons among datasets and decoding strategies, aligning with the key evaluation aspects of our research (see RQ1 and RQ2). Given the space constraints and the high dimensionality of the presented study, which involves 9 decoding strategies, 10 metrics, and 6 datasets, our goal in the main paper was to delve into relative efficiency and effectiveness comparisons. We wanted to identify patterns and assess the impact of decoding choices, rather than focus on reporting precise metric scores. However, we acknowledge the importance of clarity and transparency. As detailed in the following comments, we significantly improved our presentation style to address your concerns.
>
> **(1.a) Lack of differentiability in the chosen colors**
> > (review quote) Because of the density of information and the lack of differentiability in the chosen colors for different decoding strategies, Figure 3 is very hard to follow. For example, Greedy Search and Beam Sampling are both presented with "reddish" color, and the reviewer finds it very difficult to identify which is which in the plot.
> - Thank you for bringing this concern to our attention. In response to your observation, we revised the colors related to decoding strategies, contributing to a more accessible interpretation of our findings. We updated every figure accordingly. Additionally, we employed distinct marks (circles and triangles) to visually contrast deterministic and stochastic decoding strategies.
>
> **(1.b) Lack of tabulated data**
> > (review quote) Because of the lack of interpretability of Figure 3 and lack of tabulated data for the results, it is hard for the reviewer to confirm the quantitative differences referred to in the Results section (e.g., +48\% UNR avg. score mentioned between short and long summaries).
> - We included tabular data in Appendix G (Table 11), thus providing a detailed breakdown of Figure 3 and ensuring the reproducibility of the reported findings. Please note that the exact metric scores will be openly released in our PRISM dataset.
>
> **(1.c) Contradictions and untreated patterns**
> > (review quote) Contrary to the discussion on the relative advantage of certain decoding strategies over other in section 5.1, the impression of the reviewer from Figure 8 is that (i) sampling, top-k, and top-p sampling clearly have larger sample variance than all other strategies, even after hyperparameter tuning, on five out of six datasets; (ii) the behavior of all decoding methods changes significantly on Multi-LS dataset, showing larger scale of variation in average score and much smaller sample-specific variance; (iii) apart from the three sampling strategies mentioned in (i), the remaining methods on all datasets except Multi-LS seem to be consistently on par with each other as evaluated by most of the evaluation metrics. These observations may contradict the quantitative findings in section 5.1, and there seems to lack specific discussion on those arguably more obvious patterns than the ones presented in Figure 3 and discussed in the text.
> - We appreciate your suggestion not to overlook more evident patterns. We agree with your observations, although we found no contradictions with the quantitative findings previously reported in Section 5.1. Following your comments, we meticulously double-checked all the results outlined in the main paper and Appendix. We confirmed the content of each figure and table, except Figure 3, where we found erroneous scores. After correction, we reviewed the quantitative findings listed in Section 5, adding the suggested ones if not already present. More generally, we provided a more detailed discussion of results, enriching the answer to every research question.

---

> ### Author Response · Authors · 2023-11-23
> **Reply continuation**
>
> **(2) Lack of general recommendations**
> > (review quote) Another weakness is the generalizability of the findings. The reviewer agrees with the authors that the lack of a universal strategy for optimal decoding is substantively supported by the results, but apart from that the paper seems to be lacking in identifying other general trend or patterns in combination of the research parameters (decoding strategy, task, model, hyperparameters). This limits the value of the results to be observational rather than inferential.
> - We are dedicated to providing insights that go beyond simple observations. In line with this commitment, we revised and expanded our analysis, introducing per-dataset rankings (Appendix H) and new findings (Section 5). Combined with result verification, we identified prevalent performance patterns and established a set of recommendations to better assist practitioners in real-world summarization use cases. Following your suggestions, we attached a visual and easy-to-follow guideline to the conclusion (Appendix L), driving the selection of valuable decoding spaces (i.e., strategy + hyperparams) depending on the optimization target and AS type.
>
> **(3) Writing problems**
>
> **(3.a) Not rigorous statements**
> > (review quote) Elevating Fluency undermines Recall, Precision, and Factuality. in Section 5.2 is not a rigorous statement based on the results, what the authors have observed is a negative correlation between fluency and other quality factors as evaluated by human, but the phrasing of the finding turns such a correlation into a misleading causal statement.
> - Upon careful review, we concur with your observation that our statement could be misconstrued as implying a causal relationship. We did not intend to assert a causal link and we appreciate your diligence in bringing this to our attention. We revised the finding description by using the "negative correlation" keyphrase.
> > (review quote) It is not rigorous to draw conclusion on "factuality", "factual flaws", or "hallucination" solely based on BARTScore (e.g., Section 5.1, stochastic vs deterministic), which is at best a proxy for those quality measures. Please at the very least state in the main text that the studied automatic metrics are "proxies" of the quality dimensions of interest.
> - We added a cautionary note at the beginning of Section 5, explicitly stating that the studied automatic metrics are "proxies" for the quality dimensions of interest. We underlined this concept in other parts of the paper, including Table 3 and Appendix L.
>
> **(3.b) Typos**
> > (review quote) Typo: Table 4, shouldn't UNR be "Unique N-gram Ratio", not "unigram n-gram ratio"?
> - We amended the reported typographical errors and revised the entire paper for improved fluency and clarity.
>
> **Q1: Scoring scale in Figure 3**
> > (review quote) In Figure 3, why are some of the metrics score concentrated towards low value (center of each circle, e.g., Perplexity on Multi-News)? The reviewer may have misunderstood the presentation here, but if all scores are rescaled to [0, 1] based on min and max of each score, shouldn't there be a point on the outer perimeter for every metrics on every dataset (that is, the decoding method with the maximum score in a metric on a dataset)?
> - As explained in the caption, [0, 1] rescaling is based on min-max normalization across all 2656 runs, which is key to pinpoint inter-dataset patterns. For each decoding strategy, tens or hundreds of hyperparameter configurations are tested (see Table 2). The scores depicted in Figure 3 come from the average pooling. Therefore, having maximum (1) or minimum (0) scores on average is improbable.
>
> **Q2: Clarification on Figure 4**
> > (review quote) Any special reason why y-axis in Figure 4 is not presented on a linear scale?
> - The y-axis in Figure 4 employs a non-linear scale to highlight variations among the predominant values (1-4 seconds) while ensuring clear visibility of outliers exceeding 10 seconds. In this way, we avoid squeezing distinctions within comparable time intervals, thereby facilitating the discovery of insights and enabling well-informed decision-making.

---

> > ### Author Response · Authors · 2023-11-23
> > **Reply continuation**
> >
> > **Q3: More details on human evaluation**
> > > (review quote) Can you elaborate, maybe in the appendix, more details on the training regimen used in the human evaluation? Were the annotators given any examples or detailed explanation on each quality dimension? How did you guarantee that the annotator would view the source document? Or were the annotators already familiar with the research objectives so that training was minimal?
> > - The judges are NLP researchers with experience in abstractive summarization; we detailed this aspect in Section 4.4. During the annotation process, the annotator can always consult the source document by pressing a button (see Figure 7). Since it is requested by only one dimension out of 4 (i.e., Faithfulness), it is kept hidden by default for visual simplicity. Regarding the instructions for proper annotation, please refer to our response to _[Reviewer rLZb]_, point (2).
> >
> > **Q4: Information about the Kendall's tau agreement score**
> > > (review quote) How is Kendall's tau calculated for human evaluation results? Was it calculated between two annotators across all pair selection results, or was a ranking of all decoding strategies first determined based on all selections from one annotator, and then tau calculated between two rankings? Why are the reported values so low and even negative on some datasets? If the annotators disagree significantly in their preference, wouldn't that seriously limit the possibility of drawing any robust conclusion based on their evaluation results?
> > - We conducted our human evaluation following the best practices of the field, relying on decoding-blind judgments, shuffled pairs and summary sources (Section 4.5), and detailed instructions (Appendix F). In light of your comment on the Kendall-tau scores, calculated between two annotators on all pair selection results, we reviewed the evaluation records and found no issues. An average Kendall-tau ([-1,1]) of 0.16 indicates the highly competitive level in the quality of the summaries produced by the top-3 decoding strategies in each dataset when properly tuned, often making the preference subjective. This confirms the lack of a one-size-fits-all decoding strategy. Moreover, we want to bring to the reviewer's attention that in Multi-LexSum, where the impact of strategies is lower on the quantitative front, humans disagree more, motivating that it is the only dataset with a negative agreement score.
> >
> > **Q5: Other semantic similarity metrics**
> > > (review quote) Why not consider more semantic similarity metrics (e.g., NLI-based evaluation, see https://aclanthology.org/P19-1213.pdf for example) or NER based metrics? Apart from BertScore and BartScore, almost all other metrics used are lexical based (n-gram or character overlap). It's hard to say if the variation observed in those metrics truly reflect variation in qualities of the generated summaries (and based on Figure 8, it's hard to say if significant variation has been observed).
> > - As stated in Section 4.4, we took advantage of a panoply of metrics capturing separate attributes. These are the most commonly used metrics in the literature to gauge the quality of text summarization. Furthermore, as remarked in Appendix F, we conducted tests using SummaC, which is an NLI-based metric to evaluate factuality, but its computational demands prevented its large-scale adoption.
> >
> > Further elaboration on the introduced changes can be found in the general comment.

---

### Official Review · Reviewer_Eedp · 2023-11-01

**Soundness:** 3 good
**Presentation:** 4 excellent
**Contribution:** 3 good
**Rating:** 8
**Confidence:** 3

**Summary:**

The paper investigates in-depth different decoding strategies for abstractive summarization, exploring more than 2500 combinations of 3 models, 6 datasets, and 9 decoding settings. The goal of the paper is to shed light on the field and demonstrate that optimized decoding choices can yield substantial improvements to the performance.

**Strengths:**

* The experiments are very comprehensive, covering a bunch of different decoding strategies and their hyperparameters.

* The data when released can be very useful, not just for further benchmarking, but also in terms of modeling and evaluation since predictions are also released.

* The paper is very clearly written.

**Weaknesses:**

* While the experiments are comprehensive, I find that the paper lacked overall recommendations that practitioners can follow in a real-world setting (which I believe is the main purpose of the paper). For example, if one has a new summarization use case, how does the paper help them decide which model and which decoding strategies (+ hyperparams) to use? The paper does provide a list of findings, however a set of recommendations as part of the conclusion would greatly improve the paper's impact.

* The models are unfortunately limited to 400-500M parameters. This is significantly small if compared with LLMs, thus it is uncertain whether the results shown in this paper transfer to these large models. It could have helped to see models of varying sizes (perhaps one small-scale and one XL-scale) to show that the results to actually hold even when the model size is different.

**Questions:**

* What is the set of recommendations that the authors think practitioners should follow?

* How do these results transfer at scale (smaller or larger)?

---

> ### Author Response · Authors · 2023-11-23
> **Thank you for your review!**
>
> Thank you for your review! We appreciate your constructive feedback. It is great to learn that you enjoyed reading our paper.
>
> **(1) Lack of overall recommendations**
> > (review quote) While the experiments are comprehensive, I find that the paper lacked overall recommendations that practitioners can follow in a real-world setting (which I believe is the main purpose of the paper). For example, if one has a new summarization use case, how does the paper help them decide which model and which decoding strategies (+ hyperparams) to use? The paper does provide a list of findings, however a set of recommendations as part of the conclusion would greatly improve the paper's impact.
> - We acknowledge your concerns. In response, we revised and expanded our analysis, introducing per-dataset rankings (Appendix H) and new findings (Section 5). We identified prevalent performance patterns and established a set of recommendations to better assist practitioners in real-world summarization use cases. Following your suggestions, we attached a visual and easy-to-follow guideline to the conclusion (Appendix L), driving the selection of valuable decoding spaces (i.e., strategy + hyperparams) depending on the optimization target and AS type.
>
> **(2) Different model sizes**
> > (review quote) The models are unfortunately limited to 400-500M parameters. This is significantly small if compared with LLMs, thus it is uncertain whether the results shown in this paper transfer to these large models. It could have helped to see models of varying sizes (perhaps one small-scale and one XL-scale) to show that the results to actually hold even when the model size is different.**
> - We recognize your point. We did not experiment with LLMs and tiny models. As explicitly outlined in the abstract, introduction, related works, method, and Appendix N, the focal point of our study was the evaluation of models predominantly utilized by the AS community, as opposed to those that may rise to prominence in the future. It is important to note that, at the beginning of this project, open-source LLMs were not as widespread as they are today. Our meticulous model selection process involved identifying cutting-edge solutions within each AS family (short, long, multi-document). The chosen models exhibit an encoder-decoder architecture and fall within the range of a million parameters or barely surpass 10B---a lower boundary often cited when discussing LLMs.  To further elucidate our decision-making, we refined and expanded our bibliometric analysis within the SCOPUS database.  In Appendix B, we introduced a chart depicting the intersection of AS and LLMs in the literature. The analysis unveiled that a mere 15.8\% of conference papers from 2020 onward explicitly integrate LLMs and AS in their title, abstract, and keywords. While an extension of our study to include LLMs and tiny LMs would be commendable, focusing solely on one model for these families would be unsound, with the risk of including biased non-representative decoding-level conclusions. Consequently, it is imperative to encompass several new models. In this sense, we emphasize that the completion of our study necessitated 73 cumulative days of computation using NVIDIA 3090 RTX Turbo GPUs (Section 4.1). In particular, incorporating experiments with LLMs would entail a substantial additional commitment of time and resources. Our empirical tests revealed that transitioning from 440M models to 7B models (Bloom, Falcon, LLaMA-2) results in an average increase in decoding time per instance by a factor of x150. It becomes crucial to strike a balance between the depth of analysis and the practical constraints. As such, we believe that this extension warrants a dedicated effort and a separate study.
>
> Further elaboration on the introduced changes can be found in the general comment.

---

### Author Response · Authors · 2023-11-22
**Updates to the paper**

The authors thank all reviewers for their constructive feedback. We are about to conclude the response for each reviewer individually. We have solved all reported weaknesses and updated the paper accordingly.

The edits include:
- A revised presentation of averaged metric results in the radar plots (Figure 3). We improved readability, making the results much more interpretable. We changed the colors to better differentiate each decoding strategy. We employed distinct marks (circles and triangles) to visually contrast deterministic and stochastic decoding strategies. _[Reviewers 8Gmg, rLZb]_.
- Meticulous validation of the results outlined in the main paper and Appendix. Correction of erroneous scores found in Figure 3. _[Reviewer 8Gmg]_.
- Introduction of tabulated data for quantitative results (Table 11), ensuring the reproducibility of all reported findings. _[Reviewers 8Gmg, rLZb]_.
- Novel discussion of results (Section 5). A more precise and in-depth dissection of quantitative and qualitative findings. We reworked and enriched the answer to every research question, gaining space by relocating the table containing distributional statistics on the explored datasets to Appendix C. _[Reviewers 8Gmg, rLZb]_.
- Analysis and visualization of decoding strategy rankings with respect to paramount metrics, expanding the examination of metric score distributions (Appendix H, Figures 10, 11, 12). We granted readers a wealth of new insights into the robustness of metrics and inter-dataset patterns. _[Reviewers Eedp, 8Gmg, rLZb]_.
- Identification and disclosure of prevailing performance patterns combined with research parameters to increase the generalizability of findings and potential impact. Construction of an easy-to-follow guideline, attached to the conclusion (Appendix L), with practical recommendations depending on the optimization target and AS type. The guideline will help practitioners choose valuable decoding spaces depending on the case study at hand. _[Reviewers Eedp, 8Gmg, rLZb, BSXa]_.
- We fixed typos and revised the text of each section, including the Appendix, to enhance the presentation quality of the entire paper. We ensured the rigor and soundness of each statement, adding cautionary notes when necessary. _[Reviewers 8Gmg, rLZb]_.
- Bibliometric analysis enrichment. We added a chart in Appendix B to portray the intersection of abstractive summarization and large language models in the literature, which is only 15.8\% of the total publication count. We delved deeper into the reasons guiding our model selection (BART, LED, PRIMERA), with further details in footnote 3 and Appendix N. _[Reviewers Eedp, BSXa]_.
- Details on human evaluation. We elaborated more on the judges' expertise, Kendall's tau calculation, and established instructions for proper annotation (Appendix F). _[Reviewers 8Gmg, rLZb]_.

---

### Meta-Review · Area_Chair_s8p1 · 2023-12-06

**Metareview:**

This paper explores a wide range of settings for abstractive summarization, particularly focused on decoding strategies.  Nine decoding strategies are explored, mostly standard ones from the literature, including recent modifications to nucleus sampling like $\eta$ sampling and more sophisticated methods like contrastive search.  Six datasets are explored across three categories.  Greedy/diverse beam search/beam search do well, but there is a range of model performance and different approaches matter substantially on some metrics.  The paper releases a dataset of its generated summaries.

The reviewers praised the scope of the experimentation here. It's a very thorough paper. Furthermore, decoding methods have not been explored that extensively for this task. After revision, the paper is quite clear to read as well.

That said, the reviewers bring up several critiques of the work.  One critique is lack of concrete recommendations from the work: there are a lot of results and analysis, but the ultimate takeaways are a bit thin.  Much of the recommendation is "use greedy/beam search" which is already commonplace in summarization literature.  Other recommendations are "there's variance in the results." While true, this is not a striking result.

8Gmg brings up a point about interannotator agreement. The values reported in Figure 5 are concerning for the validity of these results.

The LMs explored are on the smaller side. I don't think this is a critical weakness, but given recent findings about the superiority of LLMs for summarization, it limits the potential future impact.

Furthermore, I would also point out this recent paper that explored decoding strategies with respect to faithfulness: https://aclanthology.org/2023.eacl-main.210.pdf

All together, when considering an empirical analysis paper like this one, it hinges heavily on the value of the results (and the dataset) for advancing the research area. While the reviews were mixed, the value and novelty of the study are ultimately somewhat thin.

**Justification For Why Not Higher Score:**

See the last line of my meta-review: I think this paper is very thorough empirically and so hard to reject for the reviewers (hence the 5s and 6s), but ultimately not quite up to the bar of other papers in my batch. The 8 reviewer doesn't say anything that really convinces me of a positive view either.

**Justification For Why Not Lower Score:**

N/A

---

### Decision · Program_Chairs · 2024-01-16

Reject